# Usable Analytical Expressions for Temperature Distribution Induced by Ultrafast Laser Pulses in Dielectric Solids

**DOI:** 10.3390/mi15020196

**Published:** 2024-01-27

**Authors:** Ruyue Que, Matthieu Lancry, Bertrand Poumellec

**Affiliations:** Institut de Chimie Moléculaire et des Matériaux d’Orsay, Université Paris-Saclay, CNRS, 91405 Orsay, France; queruyue@hotmail.com (R.Q.); matthieu.lancry@universite-paris-saclay.fr (M.L.)

**Keywords:** temperature distribution, femtosecond pulsed laser, interaction laser–dielectric solid

## Abstract

This paper focuses on the critical role of temperature in ultrafast direct laser writing processes, where temperature changes can trigger or exclusively drive certain transformations, such as phase transitions. It is important to consider both the temporal dynamics and spatial temperature distribution for the effective control of material modifications. We present analytical expressions for temperature variations induced by multi-pulse absorption, applicable to pulse durations significantly shorter than nanoseconds within a spherical energy source. The objective is to provide easy-to-use expressions to facilitate engineering tasks. Specifically, the expressions are shown to depend on just two parameters: the initial temperature at the center denoted as *T*_00_ and a factor *R_τ_* representing the ratio of the pulse period *τ_p_* to the diffusion time *τ_d_*. We show that temperature, oscillating between *T_max_* and *T_min_*, reaches a steady state and we calculate the least number of pulses required to reach the steady state. The paper defines the occurrence of heat accumulation precisely and elucidates that a temperature increase does not accompany systematically heat accumulation but depends on a set of laser parameters. It also highlights the temporal differences in temperature at the focus compared to areas outside the focus. Furthermore, the study suggests circumstances under which averaging the temperature over the pulse period can provide an even simpler approach. This work is instrumental in comprehending the diverse temperature effects observed in various experiments and in preparing for experimental setup. It also aids in determining whether temperature plays a role in the processes of direct laser writing. Toward the end of the paper, several application examples are provided.

## 1. Introduction

In the context of an ultrafast laser interacting with solids, temperature plays a special role in the transformation processes. Some of the processes can be thermally activated, others can be temperature driven, such as phase transition but not thermally activated. The objective of this paper is to develop an analytic approximation to predict the behavior of the spatial temperature distribution and the temperature evolution over time according to the key laser parameter combinations and then to deduce their importance. This approach seeks to provide physical insight and semi-quantitative results without relying on heavy and overly detailed finite element calculations. This methodology resonates with the philosophy espoused by Paul Dirac in 1929, as documented in the Proceedings of the Royal Society of London [1]:

“*The underlying physical laws necessary for the mathematical theory of a large part of physics and the whole of chemistry are thus completely known, and the difficulty is only that the exact application of these laws leads to equations much too complicated to be soluble. It therefore becomes desirable that approximate practical methods of applying quantum mechanics should be developed, which can lead to an explanation of the main features of complex atomic systems without too much computation*”.

In the ultrafast laser–matter interaction process, the energy from the laser pulse that has an extremely short pulse duration (10^−11^–10^−14^ s) is partially injected into a small focal volume of transparent dielectric solids. This intense laser pulse with high irradiance (>10^13^ Wcm^−2^) in the focal region stimulates a series of complex dynamic processes, such as multiphoton ionization, tunneling ionization, inverse bremsstrahlung absorption, and avalanche ionization within an ultrashort time scale [2]. Such interactions lead to high-density electron excitations in condensed matter, creating plasma with high temperatures and pressures. This plasma expands rapidly in the focal zone, resulting in structural modifications as energy relaxes through phonon–electron interactions [3,4].

In the low repetition rate regime, thermal accumulation is usually negligible in the processing. The temperature decreases to the initial degree before the next pulse arrives. The non-linear nature of the optical absorption can confine the formed modifications to the focal volume. These advantages minimize the thermal collateral damage and heat-affected zone [5]. Thus, ultrafast laser direct writing (ULDW) is suggested as a general technique to induce highly localized modifications and optical structures within/near the focus in a wide range of transparent solids [6,7,8,9,10,11]. In this regime, denoted as non-thermal ULDW, the repetition rate (RR) is usually a few kilohertz, and the fabrication efficiency is also limited by the low pulse RR.

In contrast, when the pulse repetition rate of the ultrafast laser increases, the interval between successive laser pulses is less than the time needed for the absorbed energy to diffuse out of the focal volume and this induces an obvious localized heat accumulation effect [11,12,13,14,15,16,17,18]. In this case, for a given pulse energy, the temperature increases continuously in the focal zone before stabilizing. The final diffusion of the heat into the surrounding material may lead to a material melting beyond the focal volume over a longer time scale. In this regime, denoted as thermal ULDW, the melted modified region is much larger than the focus size. Paralleling to the wide applications of non-thermal ULDW, the localized thermal accumulation has been demonstrated to be important in the ULDW for inducing various phenomena and structures in the transparent solids and improving the performance of the fabricated devices. For example, the thermal accumulation can lead to a higher symmetry of the waveguide cross-section, reducing the propagation loss and enhancing the fabrication efficiency [12,15,16]. Thermal accumulation in the ULDW can also induce unique phenomena, such as elemental redistribution and local crystallization, which are nearly not achievable in the non-thermal ULDW [11,19,20,21,22,23]. In the thermal ULDW regime, the temperature distribution can work as a driving force to redistribute the elements or reorganize the structures in the thermal melted region. The thermal accumulation effect has also been reported to be critical for the formation of periodic nanogratings in some glass systems [24]. Moreover, the thermal accumulation induces a high temperature that can produce thermally excited free electrons, which seeds the avalanche ionization and significantly enhance absorptivity [25]. As a result, more energy can be absorbed, and this further increases thermal accumulation. Until now, thermal accumulation has been established to be an important assistor in many cases to help ULDW to achieve various applications in fundamental science and technological manufacturing [14,15,16,21,25,26]. Clarifying the principle of thermal ULDW and reviewing its current stage in the applications are highly urgent and significant for guiding future work [11,15,16].

For this aspect of the work, Lax et al. in 1977 [27] published the first paper that described the 3D spatial distribution of the temperature rise induced by the Beer–Lambert absorption of a static Gaussian CW laser beam in cylindrical geometry. Then, Sanders in 1984 [28] described an extension of these calculations for scanning beams and provided analytic expressions. In 1991, Haba et al. [29] described the calculation of a 3D spatial distribution for the Beer–Lambert absorption of a scanning Gaussian pulsed laser in cylindrical geometry. However, even if the expression was quite complete but numerically solvable, there was no extended discussion on the laser/material parameters. Then, Eaton et al. [15] in 2005 and Zhang et al. [30] in 2007 performed finite difference calculations, for simple pulsed and CW Gaussian beams in spherical geometry, preventing easy material analysis. In 2007, Sakakura et al. [18] solved the Fourier equation in the frame of cylindrical geometry for energy delivered by a Gaussian pulsed fs laser (pulse duration 220 fs, RR 3 Hz, pulse energy < 1 μJ). With such a weak RR, the calculation can be restricted to one pulse as the experimental measurement (a lens effect) was smaller than 1 ms. However, it is not a special case and for material treatment, a large number of pulses are required. That is why Miyamoto et al. [31] in the same year, deduced analytical expressions for scanning uniform pulsed laser energy deposition in a parallelepiped volume of width 2 w corresponding to the scanning CW beam diameter at 1/*e* and length corresponding to the absorption length (1/*α*). These calculations were used also by Beresna et al. [32] and applied to a particular case, i.e., borosilicate. In 2011, Miyamoto et al. [25] considered a cylindrical source with its full width dependent on z in order to account for the convergence of the beam or the non-linear properties including the self-focusing. In 2012, Shimizu et al. [33] used a static cylindrical Gaussian beam and Gaussian energy deposition in depth for multi-pulsed laser energy deposition but solved the problem numerically. Lastly, in 2019 and 2020, Rahaman et al. [34,35] proposed an analytical solution using Duhamel’s theorem and Hankel’s transform method, for a transient, two-dimensional thermal model. We summarized the above research in Table 1 below, to compare with our work.

In short, the drawback in the available literature is that the authors did not provide simple expressions that allow the reader to easily understand how each parameter of lasers and materials influences the evolution of the temperature distribution, and to control the thermal effect in transparent materials with non-linear optical absorption for which the effect is mainly in volume for a focused beam. However, beyond this step that corresponds to the absorption of a small part of the pulse energy, the absorption becomes linear [36]. This is the reason why we present the analytical approach or link the properties of the materials to the shape of the temperature distribution and use it for explaining the phenomena such as:-The appearance of several regions in the heat-affected volume including change of the structure of a glass, crystallization, phase separation, thermal erasure while writing providing that energy endo or exo is negligible in front of the laser one;-The variations in the shape of the interaction volume according to the laser parameters like a change of laser track width, change of laser track morphology.

For this purpose, we restricted ourselves assuming that the physical properties of the material are independent of the temperature, but this does not prevent the possibility of physical deductions. We used the simplest solution of the Fourier equation in spherical geometry, i.e., a Gaussian shape along the perpendicular and longitudinal direction of the beam propagation direction. This applies not only to the sample surface but also to multiphoton absorption by stating the coordinate origin at the geometrical or effective focus. Since the typical application of this model is the thermal accumulation of a high focused beam in a material with non-linear absorption. Namely, the cylindrical symmetry and the Beer–Lambert law along z cannot be considered. We have also considered that the pulse duration (smaller than a few ns) is much smaller than the diffusion time so that the initial temperature distribution is defined by the shape of the absorbed energy source. This is applicable in most cases to femtosecond and nanosecond lasers as the diffusion time is usually of the order of a fraction of μs in inorganic glasses and a few μs in organic materials. In addition, material phase change and non-linear optical effects are not considered in this model except for the presence of coefficient A (see below).

This study was motivated by seeing nowadays that, as the means of simulation are easily accessible, the physical sense is hidden or even lost, which prevents the correct management of the laser parameters according to targeted property modifications.

## 2. Starting Formulation

From a theoretical point of view, the heat deposited at a point by the laser diffuses into the material by following Fourier’s law $\vec{q}=-\overset{̿}{\kappa }\vec{\nabla }T$ where $\vec{q}$ is the heat flow (energy per unit area and time). Fourier considers it to be linearly dependent on the temperature gradient. *κ* is the thermal conductivity, in general, a tensor of order 2 which relates the gradient vector of T to the flux. Its dimension is energy (J/m^2^·s·K). For isotropic materials, such as glasses, one will suppose that this tensor is reduced to a scalar. To calculate (in principle) the evolution and distribution of T, we start with the law of the conservation of energy, $\frac{d\rho_{Q}}{dt}+\vec{\nabla }\cdot \vec{q}=source\;terms-sink\;terms$. The source term is the laser energy density deposited per unit of time (i.e., absorbed laser power), written symbolically as $\frac{\delta \rho_{Q}}{\delta t}$. Its spatial shape defines the symmetry of the problem. For the sake of simplicity for demonstrating physical conclusions, we have assumed spherical symmetry. This means that we do not take into account some changes of focal volume with incident pulse energy due to Kerr self-focusing and electron plasma defocusing. We assume that there were no heat annihilation terms (for example, endothermic chemical transformation), sink terms=0. Using the definition of specific heat, $\frac{d\rho_{Q}}{dt}={\rho \cdot C}_{p}\cdot \frac{dT}{dt}$, ρ and $C_{p}$ are the density and specific heat capacity, respectively. $\vec{\nabla }\cdot \vec{q}=\vec{\nabla }\cdot \left(-\kappa \vec{\nabla }T\right)=-D_{th}\cdot \Delta T$, with diffusivity $D_{th}=\frac{\kappa }{{\rho \cdot C}_{p}}$, Δ is the Laplace operator written in spherical symmetry, $\Delta =\nabla^{2}=\frac{\partial^{2}}{\partial r^{2}}+\frac{2\partial }{r\partial r}$. Considering a beam moving not too fast, convection can be neglected (i.e., the time derivative of the spatial coordinate), $\frac{dT}{dt}$ is thus written as $\frac{\partial T}{\partial t}$. Therefore, we obtain the following equation:(1)$$\begin{array}{c} \frac{\partial T\left(r,t\right)}{\partial t}-D_{th}\cdot \Delta T\left(r,t\right)=\frac{1}{\rho C_{p}}\frac{\delta \rho_{Q}}{\delta t} \end{array}$$

Since the pulse duration is much less than the diffusion time (*w^2^/D_th_* with *w* is the beam waist radius at 1/*e*), the latter is at the scale of 10^−7^ s and 10^−6^ s, the diffusion process can be considered therefore to be well separated from the deposition process. During the pulse, a deposition of energy density takes place, but the diffusion does not begin, so $D_{th}\cdot \Delta T=0$, and Equation (1) becomes:(2)$$\begin{array}{c} \frac{\partial T\left(r,t\right)}{\partial t}=\frac{1}{\rho C_{p}}\frac{\delta \rho_{Q}}{\delta t} \end{array}$$

Assuming a Gaussian shape of $\frac{\delta \rho_{Q}}{\delta t}\left(r,t\right)=\frac{A\cdot E_{p}}{\pi^{\frac{3}{2}}w^{3}}\cdot exp\left[\frac{-r^{2}}{w^{2}}\right]\cdot f(t)$, where w is the beam waist radius (at 1/*e*), f(t) is the pulse shape (integral of f(t) on the pulse time = 1), $E_{p}$ is the energy of the pulse, A represents the absorbed fraction of the pulse energy. Therefore, $T\left(r,0\right)-T_{room}=\frac{1}{\rho C_{p}}\int_{pulse}\frac{\delta \rho_{Q}}{\delta t}dt=T_{00}\cdot exp\left[\frac{-r^{2}}{w^{2}}\right]$ with:(3)$$\begin{array}{c} T_{00}=\frac{A\cdot E_{p}}{\pi^{\frac{3}{2}}\rho C_{p}w^{3}} \end{array}$$

After pulse energy deposition, diffusion begins to operate, and Equation (1) becomes:(4)$$\begin{array}{c} \frac{\partial T\left(r,t\right)}{\partial t}-D_{th}\cdot \Delta T\left(r,t\right)=0 \end{array}$$

Using the initial and boundary conditions on solutions of Equations (2) and (4), we obtain:(5)$$\begin{array}{c} T\left(r,t\right)=T_{00}\cdot \frac{w^{3}}{{\left(w^{2}+4D_{th}\cdot t\right)}^{\frac{3}{2}}}\cdot exp\left[-\left(\frac{r^{2}}{w^{2}+4D_{th}\cdot t}\right)\right]+T_{room} \end{array}$$

Equation (5) describes a single-pulse-induced temperature distribution over time. $T_{00}$ is the maximum temperature induced by a laser pulse at the focus center. $T_{room}$ is the ambient temperature, which will be omitted for ease of calculation. The temperature should thus be understood as the temperature increment above the initial sample temperature.

It is important to note that when utilizing the spherical model, the deposited energy volume will consistently yield a higher temperature than reality, as the size along z is usually larger than the waist radius. Given our primary concern lies in assessing the temperature’s dependence on various parameters, it is possible to adjust the actual calculated value, which is notably affected by the absorption fraction A, to align it more closely with reality.

In the case of the absorption of *N* pulses, we easily obtain the evolution of the distribution considering the linearity of the differential equation and making up the sum of the solution for one impulsion but shifted in time $\tau_{p}=1/RR$, where *RR* is the pulse repetition rate:(6)$$\begin{array}{c} T\left(r,t\right)=T_{00}\cdot \sum_{\begin{array}{c} n=0\\ N-1=integer\;part\left(t\cdot f\right) \end{array}}^{N-1}\frac{w^{3}}{{\left(w^{2}+4D_{th}\cdot t\right)}^{3/2}}\cdot \exp\left[-\left(\frac{r^{2}}{w^{2}+4D_{th}\left(t-n{\cdot \tau }_{p}\right)}\right)\right] \end{array}$$

With $\tau_{d}=w^{2}/4D_{th},r_{w}=\frac{r}{w},$ Equation (6) reads:(7)$$\begin{array}{c} T\left(r_{w},t\right)=T_{00}\cdot \sum_{n=0}^{N-1}\frac{1}{{\left(1+\frac{t-n{\cdot \tau }_{p}}{\tau_{d}}\right)}^{3/2}}\cdot exp\left[-\left(\frac{{r_{w}}^{2}}{1+\frac{t-n{\cdot \tau }_{p}}{\tau_{d}}}\right)\right] \end{array}$$

We note that the variables involved in Equation (7) are the ratio between the period of the pulses $\tau_{p}$ and the diffusion times $\tau_{d}$, while the other laser and material parameters are involved in the amplitude $T_{00}$. Therefore, we introduce the parameter $R_{\tau }$:(8)$$\begin{array}{c} R_{\tau }=\frac{\tau_{p}}{\tau_{d}} \end{array}$$

Therefore, Equation (7) becomes:(9)$$\begin{array}{c} \frac{T\left(r_{w},t\right)}{T_{00}}=\sum_{\begin{array}{c} n=0\\ N-1=integer\;part\;\left(\frac{t}{\tau_{p}}\right) \end{array}}^{N-1}\frac{1}{{\left[1+\left(\frac{t}{\tau_{p}}-n\right)\cdot R_{\tau }\right]}^{\frac{3}{2}}}\cdot \exp\left[-\frac{{\left(r_{w}\right)}^{2}}{1+\left(\frac{t}{\tau_{p}}-n\right)\cdot R_{\tau }}\right] \end{array}$$
where N is the number of pulses defined from the time t.

The objective now is to compute the value of the temperature T according to the coordinate $r_{w}$ when *N* $\gg 1/R_{\tau }$. We will show how the temperature changes with the number of pulses according to heat accumulation (hence $R_{\tau }$), i.e., when *T* at the end of the period cumulates with the increase induced by the absorption of the next pulse. We will also describe the properties of the temperature on average in the pulse repetition period. We will also show that a steady state can be reached and give the practical number of pulses for that.

## 3. Final Temperatures at Steady State

At first, we separate the temperature problem into two cases: (1) at the center, i.e., $r_{w}=0$; (2) for general cases when $r_{w}\neq 0$ including case (1). Again, for the sake of simplicity, $T_{00}$ will be usually omitted. Therefore, the subsequent temperatures will virtually include $T_{00}$.

### 3.1. At the Center

At the center, $r_{w}=0,$ and thus, from Equation (9):(10)$$\begin{array}{c} T\left(0,t\right)=\sum_{\begin{array}{c} n=0\\ N-1=integer\;part\;\left(\frac{t}{\tau_{p}}\right) \end{array}}^{N-1}\frac{1}{{\left[1+\left(\frac{t}{\tau_{p}}-n\right)\cdot R_{\tau }\right]}^{\frac{3}{2}}} \end{array}$$

Calling $N_{t}=\frac{t}{\tau_{p}}$. The temperature evolutions over the generalized pulse number *N_t_* for several $R_{\tau }$ are shown in Figure 1.

From Figure 1 we observe that:


-$T\left(0,N_{t}\right)$ oscillates between a minimum ($T_{min}$) and a maximum ($T_{max}$) in each period between two pulses;-The oscillation amplitude ${(T}_{osc})$ seems to be the same, whatever $R_{\tau }$;-T seems to reach a steady state as $N_{t}$ becomes large (already seen in various papers [29,31,32]);-The number of pulses to reach this ‘steady state’ appears very small for a large $R_{\tau }$ but larger for small $R_{\tau }$ values. For a larger $R_{\tau }$, the temporal overlapping of temperature increase contributions from consecutive pulses is weaker, whereas it increases (heat accumulation) when $R_{\tau }$ is smaller.


#### 3.1.1. The Oscillation Amplitude *T_osc_*

We observe the oscillations of temperature on time in Figure 1 on each period. Just after the pulse energy deposition, the temperature experiences a sudden increase and then a slow decrease until the next pulse arrival. It is important to know the amplitude of the temperature oscillations ($T_{osc}$) because when $T_{osc}$ is large, at the beginning of a period, temperature may be high enough for transformation but in a short time, and at the end of the period during a long time duration, the temperature can be low, maybe achieving another transformation. The middle part could therefore often be the most active part.

##### The Limit of the Temperature Oscillation Amplitude after an Infinite Number of Pulses

The question here is: how do the oscillations evolve in time according to the pulse number N for a given diffusion time? If the period is large ($R_{\tau }$ large), we expect independent pulses and thus the amplitude will be $T_{00}$. However, when the pulse period is small ($R_{\tau }$ small), can we imagine a smaller oscillation? The next calculation provides answers.

For that purpose, we compare the difference between the maximum *T* and minimum *T* of the *N*th pulse, *T_max_*(0,*N*) − *T_min_*(0,*N*) = *T*(0, *t_N_*) − *T*(0, t*_N_*_+1_ − *ε*), where *ε* is an arbitrary small quantity for ensuring that the number of pulses in the expression (11) is the same. *T_max_* is defined just after the deposition of the *N*th pulse, so at the beginning of the pulse, *t_N_* = (N − 1)*τ_p_*. *T_min_* is at the end of the pulse period, just before the (*N* + 1)th pulse arrival. Using Equation (10), we have:(11)$$\begin{array}{c} T_{max}\left(0,N\right)=T\left(0,{t=t}_{N}=\left(N-1\right)\tau_{p}\right)=\sum_{n=0}^{N-1}\frac{1}{{\left[1+\left(N-1-n\right)\cdot R_{\tau }\right]}^{\frac{3}{2}}}\\ =\sum_{n^{\prime }=0}^{N-1}\frac{1}{{\left[1+n^{\prime }\cdot R_{\tau }\right]}^{\frac{3}{2}}} \end{array}$$

*T_min_* will be at *t = N*·*τ_p_ − ε,* thus not containing the temperature contribution induced by the (*N* + 1)th pulse, so:(12)$$\begin{array}{c} T_{min}\left(0,N\right)=T\left(0,{t=t}_{N+1}-\varepsilon =N\cdot \tau_{p}-\varepsilon \right)=\sum_{n=0}^{N-1}\frac{1}{{\left[1+\left(N-n\right)\cdot R_{\tau }\right]}^{\frac{3}{2}}}\\ =\sum_{n^{\prime }=1}^{N}\frac{1}{{\left[1+n^{\prime }\cdot R_{\tau }\right]}^{\frac{3}{2}}}\; \end{array}$$

Therefore,
(13)$$\begin{array}{c} T_{osc}\left(0,N\right)=T_{max}\left(0,N\right)-T_{min}\left(0,N\right)=1-\frac{1}{{\left[1+N\cdot R_{\tau }\right]}^{\frac{3}{2}}} \end{array}$$
and
(14)$$\begin{array}{c} T_{osc}\left(0,\infty \right)=\underset{N\to \infty }{\lim}T_{osc}\left(0,N\right)=1 \end{array}$$

When $N\gg 1/R_{\tau }$, $T_{osc}$ tends to 1. This means *T*_00_ in the absolute scale. $T_{osc}$ according to the pulse number is shown in Figure 2. It reaches a maximum value 1, i.e., $T_{00}$, when $N\gg 1/R_{\tau }$. At the beginning of the irradiation, $T_{osc}$ starts with a value smaller than $T_{00}$, where a smaller $R_{\tau }$ leads to a smaller oscillation at the beginning. When the pulse number N increases until some value, $T_{osc}$ reaches $T_{00}$. When $R_{\tau }$ is large, e.g., 10, the amplitude is equal to $T_{00}$ whatever N, as pulse contributions are separated (no overlapping). With the expression of $T_{00}$ (Equation (3)), which is proportional to pulse energy (*E_p_*), the temperature oscillation range can be determined.

##### The Effective Number of Pulses for Reaching the Limit of *T_osc_*($N_{sso}^{0}$)

When will the temperature in the material reach a stable oscillation? In practice, we can calculate a real number of pulses ($N_{sso}^{0}$) to closely reach the oscillation limit. (In notation, $N_{sso}^{0}$, where *sso* means steady state of oscillation, 0 means the situation when *r_w_* = 0).

Consider when $T_{osc}\left(0,N_{sso}^{0}\right)=(1-\varepsilon )\cdot T_{osc}(0,\infty )$, the limit is practically reached, where ε is a small quantity, i.e., a few % (based on the actual situation). Thus, it is:(15)$$\begin{array}{c} \frac{\left|T_{ocs}(0,\infty )-T_{osc}(0,N_{sso}^{0})\right|}{T_{ocs}(0,\infty )}<\varepsilon \; \end{array}$$

Then, we have:(16)$$\begin{array}{c} N_{sso}^{0}=\frac{1}{R_{\tau }}\left[{\left(\frac{1}{\varepsilon }\right)}^{\frac{2}{3}}-1\right] \end{array}$$

The plot of $N_{sso}^{0}$ is shown in Figure 3, note that the actual number of pulses is the integer part above 1 (as *ε* should be smaller, bounded by $\varepsilon =\frac{1}{{\left[1+R_{\tau }\right]}^{\frac{2}{3}}}$). Some specific parameters are given below for visualizing this value. With *ε* = 3%, when $R_{\tau }=1$ (conceivable combinations of material parameters and laser RR), $N_{sso}^{0}=9.36$. So, after 10 pulses, the amplitude of the oscillating temperature reaches 0.97 *T*_00_. When $R_{\tau }$ is large, e.g., $R_{\tau }=10$, $N_{sso}^{0}=0.94$, so only one pulse rules the oscillation amplitude, and we can understand that pulse contributions are separated. When $R_{\tau }$ is smaller, $N_{sso}^{o}$ increases rapidly, e.g., $R_{\tau }=0.1$, i.e., ${1/R}_{\tau }=10$, $N_{sso}^{o}=94$. Beyond $N_{sso}^{o}$ pulses, the oscillation amplitude becomes almost constant. According to the pulse period, we can know the time to reach the constant oscillation amplitude. By comparing with pulse number *N* = 1 (blue dashed line in Figure 3), we can deduce in what condition ($R_{\tau }$ larger than which value) the temperature oscillation is constant since the first pulse.

To conclude on this point, Equations (13) and (14) provides that the temperature oscillation amplitude is *T*_00_ after *N_sso_* pulses and at an oscillatory steady state. Equation (16) provides the effective number of pulses for reaching it. It takes more time when $R_{\tau }$ is very small (slow heat diffusion or high pulse *RR*) but the requested time remains quite small. In particular, the temperature oscillation amplitude is only relevant to certain laser parameters of a single pulse and material parameters of the energy-to-temperature conversion relationship, independent of RR and diffusion parameters.

##### *T_min_* and *T_max_*

We demonstrate in Appendix B that the temperature induced by laser pulses will not increase indefinitely but converge to a finite value. This defines a steady state that corresponds to the equilibrium between the energy supplied by the laser and the energy diffusing out of the irradiated voxel.

##### The Limit of *T_max_* and *T_min_*

*T_min_*: The analytical expression of the minimum temperature is transformed from the sum expression Equation (12), with details found in Appendix C. Therefore, we obtain:(17)$$\begin{array}{c} T_{min}\left(0,N\right)\approx \frac{1}{{2\left(1+R_{\tau }\right)}^{\frac{3}{2}}}+\frac{1}{2{\left(1+N\cdot R_{\tau }\right)}^{\frac{3}{2}}}+\frac{2}{R_{\tau }}\left[\frac{1}{\sqrt{1+R_{\tau }}}-\frac{1}{\sqrt{1+N\cdot R_{\tau }}}\right] \end{array}$$

This expression shows the increase of $T_{min}$ according to N and $R_{\tau }$. It is plotted in Figure 4.

The final limit of $T_{min}$, i.e., when $N\gg 1/R_{\tau }$ is given below:(18)$$\begin{array}{c} T_{min}\left(0,\infty \right)=\frac{1}{{2\left(1+R_{\tau }\right)}^{\frac{3}{2}}}+\frac{2}{R_{\tau }}\frac{1}{\sqrt{1+R_{\tau }}}\approx \frac{2}{R_{\tau }}\frac{1}{\sqrt{1+R_{\tau }}} \end{array}$$

The same method has been applied to obtain the $T_{max}$ limit, and the detail can be also found in Appendix C. We have thus:(19)$$\begin{array}{c} T_{max}(0,N)\approx 1+\frac{1}{2{\left[1+R_{\tau }\right]}^{3/2}}+\frac{1}{2{\left[1+(N-1)R_{\tau }\right]}^{3/2}}+\frac{2}{R_{\tau }}\left[\frac{1}{\sqrt{\left[1+R_{\tau }\right]}}-\frac{1}{\sqrt{1+(N-1)R_{\tau }}}\right] \end{array}$$

When $N\gg 1/R_{\tau }$, the $T_{max}$ limit is:(20)$$\begin{array}{c} T_{max}\left(0,\infty \right)=1+\frac{1}{{2\left(1+R_{\tau }\right)}^{\frac{3}{2}}}+\frac{2}{R_{\tau }}\frac{1}{\sqrt{1+R_{\tau }}}\approx 1+\frac{2}{R_{\tau }}\frac{1}{\sqrt{1+R_{\tau }}} \end{array}$$

From these expressions, we see that the difference between $T_{max}\left(0,\infty \right)$ and $T_{min}\left(0,\infty \right)$ is 1, which is consistent with the oscillation amplitude limitation (Equation (14)).

When $R_{\tau }$ reaches 0 (e.g., by increasing pulse RR or with the material of small thermal conductivity), Equations (18) and (20) are approximately proportional to $\frac{2}{R_{\tau }}$. Reintroducing here exceptionally $T_{00}$ (Equation (3)), we obtain:(21)$$\begin{array}{c} {T_{max}\left(0,\infty \right)\sim T}_{min}\left(0,\infty \right)\sim T_{00}\cdot \frac{2}{R_{\tau }}=\frac{2{AE}_{p}}{\pi^{\frac{3}{2}}\rho C_{p}w^{3}R_{\tau }}=\frac{2{AE}_{p}\tau_{d}}{\pi^{\frac{3}{2}}\rho C_{p}w^{3}\tau_{p}}=\frac{2AE_{p}f}{\pi^{\frac{3}{2}}D_{th}\rho C_{p}w}=\frac{2AP}{\pi^{\frac{3}{2}}\kappa w} \end{array}$$
with P being the average power.

We note that the temperature is now dependent on the incident laser power as is the case for CW lasers, and inversely dependent on the thermal conductivity (*κ*), whereas *T*_00_ was dependent on the incident pulse energy (*E_p_*), not on the thermal diffusivity but just on the heat capacity of the material. This is due to large time-overlapping of the pulse contribution when $R_{\tau }$ reaches 0.

Therefore, increasing the pulse *RR* with constant *E_p_* leads to a faster temperature increase but NOT with constant average power. The same maximal temperature can be achieved with or without heat accumulation. However, *T_min_*, which is negligible in front of *T_max_* for large $R_{\tau }$ values, increases until it almost equals *T_max_* for small $R_{\tau }$ values (large *RR*).

##### The Effective Number of Pulses for Reaching the Limit of *T_min_* and *T_max_* ($N_{ssmin}^{0},N_{ssmax}^{0})$

*N_ssmin_*: The effective number of pulses to reach the steady state *N_ss_* is defined to have temperature reaching $T_{min}$ or $T_{max}$. As the same definition as for *N_sso_*, the first approximation of *N_ssmin_* is obtained by solving the following assertion, $\frac{\left|T_{min}\left(0,N\right)-T_{min}\left(0,\infty \right)\right|}{T_{min}\left(0,\infty \right)}<\varepsilon$, with ε being a small quantity. Posing $X=\frac{1}{\sqrt{1+N\cdot R_{\tau }}}$, it reads $\frac{X^{3}}{2}-\frac{2}{R_{\tau }}X+\varepsilon \cdot T_{min}\left(0,\infty \right)>0$. This cubic equation has three roots, where the physical one is $X<\frac{\varepsilon \cdot T_{min}\left(0,\infty \right)}{\frac{2}{R_{\tau }}}$. Therefore,
(22)$$\begin{array}{c} N_{ssmin}^{0}>\frac{1}{R_{\tau }}\left[{\left(\frac{2}{R_{\tau }\cdot \varepsilon \cdot T_{min}\left(0,\infty \right)}\right)}^{2}-1\right] \end{array}$$

*N_ssmax_*: With the same method, we obtain:(23)$$\begin{array}{c} N_{ssmax}^{0}>\frac{1}{R_{\tau }}\left[{\left(\frac{2}{R_{\tau }\cdot \varepsilon \cdot T_{max}\left(0,\infty \right)}\right)}^{2}-1\right] \end{array}$$

The *N_ss_* for reaching closely the steady state (with *ε* departure). *T_osc_*(0,∞), *T_min_*(0,∞), and *T_max_*(0,∞) are plotted in Figure 5 according to $R_{\tau }$ (with ε=3%).

From Figure 5, we can see that with ${1/R}_{\tau }$ increasing, the effective pulse numbers for reaching the steady state increases whether for *T_osc_*, *T_min_*, or *T_max_*. For practical use, it is better to define one *N_ss_* for calculation. When $R_{\tau }$ is large, since the value of *T_min_* is almost 0 (no pulse superimposition), it is therefore not meaningful to take it into consideration. For *N_sso_* and *N_ssmax_*, the value converges to 0 when $R_{\tau }$ is large because it can be considered to be already at steady state when the pulses are separated. As observed, the green dash is always higher than the red line when *N* > 1, and the oscillation reaches the steady state faster than *T_max_*. Therefore, $N_{ssmax}^{0}$ is the suitable and practical number of pulses needed for reaching the steady state. Some examples for the *N_ssmax_* value are shown in Table 2 (with *ε* = 0.03 for organic materials and *ε* = 0.06 for inorganic materials):

We observe that for the inorganic material examples in the table, with *RR* = 200 kHz, there is no heat accumulation, pulse contributions are separated, there is no transient time, and the time variation of *T* is from one pulse contribution. However, for the organic material examples, except for glycine crystal, with same *RR*, the laser induces heat accumulation. Therefore, not only can we deduce from the known laser and material parameters whether or not there will be heat accumulation, but we can also easily backtrack on how to choose a laser *RR* that avoids or guarantees heat accumulation in a particular material.

We can thus define the boundary between the two domains by *N_ssmax_*(*R_τ_*,*ε*) = 1 as shown in Figure 5 by the blue point. When the effective pulse number is equal to 1, the pulse contribution is separated, so it is considered that there is no heat accumulation. This is the heat accumulation definition we propose with a new perspective. *R_τ_* varies with the level of sensitivity of the targeted transformation, for *ε* = 0.03, *R_τ_* = 15.7, for *ε* = 0.06, *R_τ_* = 7, for instance.

From *N_ssmax_* together with the laser pulse *RR*, we know the time needed to reach the steady state. Accordingly, the time for reaching the steady state$\;t_{ss}^{0}$ is (considering the effective number to reach the *T_max_* limit):(24)$$\begin{array}{c} t_{ss}^{0}=N_{ssmax}^{0}\tau_{p}=\tau_{d}\left[{\left(\frac{2}{R_{\tau }\cdot \varepsilon \cdot T_{max}\left(0,\infty \right)}\right)}^{2}-1\right] \end{array}$$

Figure 6 show the plots versus $\frac{1}{R_{\tau }}=\frac{RR\cdot w^{2}}{4\cdot D_{th}}$ according to three different diffusion times: 0.28 µs (silica, glycine), 1.63 µs for nifedipine, and 4.9 µs for sucrose.

For small enough values of $R_{\tau }$, the time reaches the value $\tau_{d}/\varepsilon^{2}$, i.e., 1111$\tau_{d}$ for *ε* = 3%. Note that for inorganic glass and glycine crystal cited above with $\tau_{d}$ = 0.28 µs, this time is smaller than 1 ms (red profile). However, for sucrose and nifedipine, this time is 5.5 ms and 1.8 ms, respectively. In any case, the important fact is the independency of the transient time with *R_τ_* for small values (see for *R_τ_* < 1) and thus it is bounded to quite a small value.

For large enough values of $R_{\tau }$, this time is limited by the period $\tau_{p}$ that increases with *R_τ_*.

### 3.2. Time Behavior out of the Center ($r_{w}=r/w\neq 0$)

When $r_{w}$ ≠ 0, we come back to the expression Equation (10):$$T\left(r_{w},t\right)=\sum_{\begin{array}{c} n=0\\ N-1=integer\;part\;(t/\tau_{p}) \end{array}}^{N-1}\frac{1}{{\left[1+(\frac{t}{\tau_{p}}-n)\cdot R_{\tau }\right]}^{3/2}}\cdot \exp\left[-\frac{{\left(r_{w}\right)}^{2}}{1+\left(\frac{t}{\tau_{p}}-n\right)\cdot R_{\tau }}\right]$$

Figure 7 shows the temperature evolution based on the above expression over time at two relative distances $r_{w}$ = 1 (Figure 7a,b) and $r_{w}$ = 2 (Figure 7c,d).

We observe the following differences according to radius *r_w_* = 0, 1, 2:-The amplitude of oscillation is less than 1 (in the unit of *T*_00_) for increasing radius;-The maximum temperature during a period is still at the beginning of the pulse deposition for $r_{w}$ = 1 with these three $R_{\tau }$, while at $r_{w}$ = 2 the maximum temperature is no more at the beginning. That is because there is time for heat to diffuse from the center to $r_{w}$. This renders the following calculation of *T_max_* for increasing radius to be more complex.

#### 3.2.1. *T_osc_*, *T_min_*, and *T_max_*

##### The Limit of *T_max_* and *T_min_* (when $N\gg 1/R_{\tau }$)

To calculate the oscillation amplitude *T^r^_osc_*, it is the same as the case of *r_w_* = 0. In general, we compare the difference between the maximum *T* and minimum *T* in the *N*th pulse period. *T_min_* is still considered at the end of the *N*th one, i.e., when $t=N\cdot \tau_{p}$ before the absorption of the (*N* + 1)th pulse, so:(25)$$\begin{array}{c} T_{min}\left(r_{w},N\right)=\sum_{n=0}^{N-1}\frac{1}{{\left[1+\left(N-n\right)\cdot R_{\tau }\right]}^{\frac{3}{2}}}\exp\left[-\frac{{\left(r_{w}\right)}^{2}}{1+\left(N-n\right)\cdot R_{\tau }}\right] \end{array}$$

However, since *T_max_* in some situations can be in the middle of the pulse period, we set $x_{m}$, $0\leq x_{m}\leq 1$ to define the place where the *T_max_* is. Therefore, *T_max_* is at the time $t=(N-1+x_{m})\tau_{p}$:(26)$$\begin{array}{c} T_{max}\left(r_{w},N\right)=\sum_{n=0}^{N-1}\frac{1}{{\left[1+\left(N-1+x_{m}-n\right)\cdot R_{\tau }\right]}^{\frac{3}{2}}}\exp\left[-\frac{{\left(r_{w}\right)}^{2}}{1+\left(N-1+x_{m}-n\right)\cdot R_{\tau }}\right] \end{array}$$

This expression is also a general expression to describe both *T_max_* and *T_min_*, while *T_min_* appears at the end of the period, i.e., $x_{m}$ = 1, as well as the case when *r_w_* = 0, *T_max_* appears at the beginning of the period with $x_{m}$ = 0.

The position of the maximum $x_{m}$ is solved as a function of $R_{\tau }$, $r_{w}$, and *N*. When considering the steady state, when $N\gg 1/R_{\tau }$, $x_{m}$ is shown below, and the results of the cumbersome calculation details can be found in Appendix D:(27)$$\begin{array}{c} x_{m}=\frac{\sqrt{R_{\tau }}\sqrt{9R_{\tau }+32{\left(r_{w}\right)}^{2}}-3R_{\tau }-8}{8R_{\tau }} \end{array}$$

Based on Equation (27), when considering different values of $r_{w}$ and$\;R_{\tau },$ the thermal calculation can be divided into two situations (see Appendix D):

Situation 1: $x_{m}=0$, when, *r_w_* <$\sqrt{\frac{3}{2}+\frac{2}{R_{\tau }}}$ whatever $R_{\tau }$ or $R_{\tau }$ small enough (less than $\frac{2}{{r_{w}}^{2}-1.5}$ when ${r_{w}}^{2}>1.5$).

Situation 2: $x_{m}\neq 0$, i.e., *r_w_* >$\sqrt{\frac{3}{2}+\frac{2}{R_{\tau }}}$, in this situation the maximum temperature is in the middle of the period, and the expressions of *T_max_* and *T_osc_* should contain $x_{m}$.

(1) For situation 1, when *x_m_* = 0, the limit of *T_osc_* is described as:(28)$$\begin{array}{c} T_{osc}\left(r_{w},N\right)=T_{max}\left(r_{w},N\right)-T_{min}\left(r_{w},N\right)=\exp\left[-{\left(r_{w}\right)}^{2}\right]-\frac{\exp\left[-\frac{{\left(r_{w}\right)}^{2}}{1+N\cdot R_{\tau }}\right]}{{\left(1+N\cdot R_{\tau }\right)}^{\frac{3}{2}}}\underset{N\gg 1/R_{\tau }}{\to }\exp\left[-{\left(r_{w}\right)}^{2}\right] \end{array}$$

The amplitude of the temperature oscillations reaches $T_{00}\cdot \exp\left[-{\left(r_{w}\right)}^{2}\right]$ whatever $R_{\tau }$. It is consistent with our observations in Figure 7, e.g., the amplitudes are 0.368 $T_{00}$ and 0.018 $T_{00}$ at $r_{w}=1$ and $r_{w}=2$, respectively.

Therefore, for *T_min_* and *T_max_*, using the trapezoidal rule for approximation as for *r_w_* = 0, we have *T_min_* from Equation (25):$$T_{min}(r_{w},N)\approx \frac{1}{2}\left[\frac{\exp\left[-\frac{{\left(r_{w}\right)}^{2}}{1+R_{\tau }}\right]}{{\left(1+R_{\tau }\right)}^{3/2}}+\frac{\exp\left[-\frac{{\left(r_{w}\right)}^{2}}{1+N\cdot R_{\tau }}\right]}{{\left(1+N\cdot R_{\tau }\right)}^{3/2}}\right]+\frac{\sqrt{\pi }}{R_{\tau }\cdot r_{w}}\left[erf\left(\frac{r_{w}}{\sqrt{1+R_{\tau }}}\right)-erf\left(\frac{r_{w}}{\sqrt{1+N\cdot R_{\tau }}}\right)\right]$$
(29)$$T_{min}\left(r_{w},N\;\right)\underset{N\gg 1/R_{\tau }}{\to }T_{min}(r_{w},\infty )=\frac{\exp\left[-\frac{{\left(r_{w}\right)}^{2}}{1+R_{\tau }}\right]}{{2\left(1+R_{\tau }\right)}^{3/2}}+\frac{\sqrt{\pi }}{R_{\tau }\cdot r_{w}}erf\left(\frac{r_{w}}{\sqrt{1+R_{\tau }}}\right)$$

$\frac{\exp\left[-\frac{{\left(r_{w}\right)}^{2}}{1+R_{\tau }}\right]}{{2\left(1+R_{\tau }\right)}^{3/2}}$ is called part 1, and $\frac{\sqrt{\pi }}{R_{\tau }\cdot r_{w}}erf\left(\frac{r_{w}}{\sqrt{1+R_{\tau }}}\right)$ is called part 2 for further use. *T_max_* is *T_min_ + T_osc_*.

(2) For situation 2, even if $x_{m}$ influences the *T_max_* and *T_osc_*, its influence is bounded. When $x_{m}=0$ is used, we calculate the temperature at the beginning of the period and the maximum is thus larger (with a non-zero $x_{m}$). However, how much larger? In which situations should we care about it? From the analysis, the details are described in Appendix D, we found that the difference appears only around $r_{w}=1.6$ to 4 when $R_{\tau }$ is large.

*T_min_* is the same as situation 1. For *T_max_*, using the trapezoidal rule for approximation as for *r_w_* ≠ 0, we have *T_max_* from Equation (26):$$T_{max}\left(r_{w},N,x_{m}\right)\approx \frac{\exp\left[-\frac{{\left(r_{w}\right)}^{2}}{1+x_{m}\cdot R_{\tau }}\right]}{{\left[1+x_{m}\cdot R_{\tau }\right]}^{\frac{3}{2}}}+\frac{\exp\left[-\frac{{\left(r_{w}\right)}^{2}}{1+{(1+x}_{m})\cdot R_{\tau }}\right]}{2{\left[1+\left(1+x_{m}\right)\cdot R_{\tau }\right]}^{\frac{3}{2}}}+\frac{\exp\left[-\frac{{\left(r_{w}\right)}^{2}}{1+\left(N-1+x_{m}\right)\cdot R_{\tau }}\right]}{{2\left[1+\left(N-1+x_{m}\right)\cdot R_{\tau }\right]}^{\frac{3}{2}}}\;+\frac{\sqrt{\pi }}{R_{\tau }\cdot r_{w}}\left\{erf\left[\frac{r_{w}}{\sqrt{\left[1+{(1+x}_{m})\cdot R_{\tau }\right]}}\right]-erf\left[\frac{r_{w}}{\sqrt{\left[1+\left(N-1+x_{m}\right)\cdot R_{\tau }\right]}}\right]\right\}$$
(30)$$\underset{N\gg 1/R_{\tau }}{\to }\;\frac{\exp\left[-\frac{{\left(r_{w}\right)}^{2}}{1+x_{m}\cdot R_{\tau }}\right]}{{\left[1+x_{m}\cdot R_{\tau }\right]}^{\frac{3}{2}}}+\frac{\exp\left[-\frac{{\left(r_{w}\right)}^{2}}{1+{(1+x}_{m})\cdot R_{\tau }}\right]}{2{\left[1+\left(1+x_{m}\right)\cdot R_{\tau }\right]}^{\frac{3}{2}}}+\frac{\sqrt{\pi }}{R_{\tau }\cdot r_{w}}\left\{erf\left[\frac{r_{w}}{\sqrt{\left[1+{(1+x}_{m})\cdot R_{\tau }\right]}}\right]\right\}$$

$\frac{exp\left[-\frac{{\left(r_{w}\right)}^{2}}{1+x_{m}\cdot R_{\tau }}\right]}{{\left[1+x_{m}\cdot R_{\tau }\right]}^{3/2}},\;\frac{exp\left[-\frac{{\left(r_{w}\right)}^{2}}{1+{(1+x}_{m})\cdot R_{\tau }}\right]}{2{\left[1+(1+x_{m})\cdot R_{\tau }\right]}^{3/2}},\;\frac{\sqrt{\pi }}{R_{\tau }\cdot r_{w}}\left\{erf\left[\frac{r_{w}}{\sqrt{\left[1+{(1+x}_{m})\cdot R_{\tau }\right]}}\right]\right\}$ are called part 1, 2, and 3, respectively.

The general expression of *T_osc_* (when *N* tends to infinity or larger than the effective number for reaching the steady state) is given as $T_{max}$ (Equation (30)) minus $T_{min}$ (Equation (29)), and it reads:(31)$$\begin{array}{c} T_{osc}\left(R_{\tau },r_{w}\right)=\frac{\exp\left[-\frac{{\left(r_{w}\right)}^{2}}{1+x_{m}\cdot R_{\tau }}\right]}{{\left[1+x_{m}\cdot R_{\tau }\right]}^{\frac{3}{2}}}+\frac{\exp\left[-\frac{{\left(r_{w}\right)}^{2}}{1+{(1+x}_{m})\cdot R_{\tau }}\right]}{2{\left[1+\left(1+x_{m}\right)\cdot R_{\tau }\right]}^{\frac{3}{2}}}+\frac{\sqrt{\pi }}{R_{\tau }\cdot r_{w}}\left\{erf\left[\frac{r_{w}}{\sqrt{\left[1+{(1+x}_{m})\cdot R_{\tau }\right]}}\right]\right\}\\ -\frac{\exp\left[-\frac{{\left(r_{w}\right)}^{2}}{1+R_{\tau }}\right]}{{2\left(1+R_{\tau }\right)}^{\frac{3}{2}}}-\frac{\sqrt{\pi }}{R_{\tau }\cdot r_{w}}erf\left(\frac{r_{w}}{\sqrt{1+R_{\tau }}}\right) \end{array}$$

Part 1 in Equation (29) and part 2 in Equation (30) are smaller than the other parts by a factor 10, so they can be approximately omitted to simplify the expressions in practice.
(32)$$\begin{array}{c} T_{osc}\left(R_{\tau },r_{w}\right)\approx \;\frac{\exp\left[-\frac{{\left(r_{w}\right)}^{2}}{1+x_{m}\cdot R_{\tau }}\right]}{{\left[1+x_{m}\cdot R_{\tau }\right]}^{\frac{3}{2}}}+\frac{\sqrt{\pi }}{R_{\tau }\cdot r_{w}}\left\{erf\left[\frac{r_{w}}{\sqrt{\left[1+{(1+x}_{m})\cdot R_{\tau }\right]}}\right]\right\}-\frac{\sqrt{\pi }}{R_{\tau }\cdot r_{w}}erf\left(\frac{r_{w}}{\sqrt{1+R_{\tau }}}\right) \end{array}$$

It is worth noticing (see Appendix D, Figure A4a,b) that when $R_{\tau }$ increases, *T_osc_* exhibits a small departure from the exact value at around *r_w_* = 2, attributed to the existence of a non-zero *x_m_*. This departure, if it is generally not negligible, is nevertheless bounded. It is calculated to be $\;\frac{1}{2.44{\left(r_{w}\right)}^{3}}$ for a large $R_{\tau }$ (details are shown in Appendix D Figure A4c,d by plotting *T_osc_* according to *r_w_* and $R_{\tau }$). Therefore, the range of *T_osc_* is given by Equations (33) and (34):(33)$$\begin{array}{c} T_{osc}\left(R_{\tau },r_{w}\right)\overset{R_{\tau }\to 0\;}{\to }\exp\left[-{\left(r_{w}\right)}^{2}\right] \end{array}$$
(34)$$\begin{array}{c} T_{osc}\left(R_{\tau },r_{w}\right)\overset{R_{\tau }\to \infty \;}{\to }\frac{1}{2.44{\left(r_{w}\right)}^{3}} \end{array}$$

We note that the oscillation amplitude *T_osc_* at situation 1 is $\exp\left[-{\left(r_{w}\right)}^{2}\right]$ which is the minimum, while in situation 2, the amplitude is larger due to the influence of *x_m_*, with a maximum value of $\frac{1}{2.44{\left(r_{w}\right)}^{3}}$ at the place around *r_w_* = 2. By now, with these temperature expressions, we obtain the spatial distribution of the minimum and maximum temperature for a given $R_{\tau }$ at steady state, shown in Figure 8. The temperature is oscillating between these two temperature profiles, and note that at *r_w_* = 0, the difference is always 1 regardless of $R_{\tau }$.

We have now all the information for plotting the *T* distribution with any *R_τ_* value.

From Figure 8, we can see that when $R_{\tau }$ is small (large frequency or small diffusivity), *T_min_* and *T_max_* have no large relative difference compared to their average values because the oscillation amplitude is always limited to $\exp\left[-{\left(r_{w}\right)}^{2}\right]$ whereas the Tmean amplitude is converging to 2/$R_{\tau }$ (heat accumulation). This case is interesting if a rather stable temperature is requested. Then, pulse energy can be adjusted for compensating the pulse RR increase. It is also worth noting that, by decreasing $R_{\tau }$, the shape of the curve converges to the *erf*(*r_w_*) curve and decreases much slower than a Gaussian one.

For large values of $R_{\tau }$ (small frequency or large diffusivity), the oscillations are relatively large as the pulses are separated and thus *T_min_* appears to have small values. It is negligible (<3%) when $R_{\tau }>16$ or <6% for $R_{\tau }>7$). This limits the domain of heat accumulation. The calculation shows that the shape of *T_max_* is also converging with increasing $R_{\tau }$ to the shape of the beam energy distribution (Gaussian, here $\exp\left[-{\left(r_{w}\right)}^{2}\right]$) independent of $R_{\tau }$. This translates that the maximum is almost whatever the radius, at the beginning of the period. This is not true exactly only around *r* = 2*w* where a few % departure from the Gaussian shape of *T_max_* is demonstrated.

For $R_{\tau }$ intermediate values, the temperature oscillations are limited between *T_max_* and *T_min_*. This is shown with particular cases with a shoe box in [31] for $R_{\tau }=2\;and\;20$ or in [32] for $R_{\tau }=20$.

However, in this paper, we regard that when *r_w_* > 2, the difference between *T_min_* and *T_max_* is vanishing. We see this in [37].

Therefore, we can deduce in particular, in whatever situation of *r_w_* and $R_{\tau }$, the temperature oscillations can be neglected and the use of an average temperature is applicable.

Other application remarks:


(1)In the intermediate cases around *R_τ_* = 1, the center of the heat-affected zone experiences large temperature oscillations whereas the periphery temperature is not oscillating. This may induce differences in the modification structures along the radius. Specifically, the pedestal of the curve, borne by *T_min_*, increases in width with $R_{\tau }$ as $1.75+\frac{1}{16}R_{\tau }\frac{R_{\tau }+145}{R_{\tau }+20}$;(2)For smaller $R_{\tau }$ values, during the transient period (before *N_ss_*), the width of the temperature distribution starts with the beam waist (Gaussian) and then increases until a size which is defined by $R_{\tau }$. It does not increase indefinitely over time. The order of magnitude is one w per two orders of magnitude on $R_{\tau }$, e.g., the trace width at 1 MHz is twice the one at 10 kHz.


##### The Effective Number of Pulses for Reaching the Temperature Limits

Since $x_{m}$ is not negligible in very limited circumstances, the effective numbers of pulses for reaching the limit of *T_osc_*, *T_min_*, and *T_max_(N^r^_sso/ssmin/ssmax_)* are given in the situation when $x_{m}=0$.

With the same definition as we calculated in $r_{w}=0$, with ε being a small quantity and $X=\frac{1}{\sqrt{1+N\cdot R\tau }}$, we have $\frac{\left|T_{osc/max/min}\left(r_{w},N\right)-T_{osc/max/min}\left(r_{w},\infty \right)\right|}{T_{osc/max/min}\left(r_{w},\infty \right)}<\varepsilon$. Therefore, the *N^r^_sso/ssmin/ssmax_* solutions are shown below:(35)$$\begin{array}{c} N_{sso}^{r}>\frac{-3\cdot W\left[-\frac{1}{3}e^{-\frac{1}{3}{r_{w}}^{2}}\cdot {r_{w}}^{2}\cdot \varepsilon^{\frac{2}{3}}\right]-2{r_{w}}^{2}}{3\cdot R_{\tau }\cdot W\left[\left(-\frac{1}{3}e^{-\frac{1}{3}{r_{w}}^{2}}\cdot {r_{w}}^{2}\cdot \varepsilon^{\frac{2}{3}}\right)\right]}=\frac{1}{R_{\tau }}\left[-\frac{2{r_{w}}^{2}}{3\cdot W\left[\left(-\frac{1}{3}e^{-\frac{1}{3}{r_{w}}^{2}}\cdot {r_{w}}^{2}\cdot \varepsilon^{\frac{2}{3}}\right)\right]}-1\right] \end{array}$$

This expression does not have an analytic root without using the tabulated function *W*, i.e., the Lambert W function (defined as $\omega e^{\omega }=z,\;\omega =W\left(z\right)$). In practice, since $X^{2}\ll 1$, by approximation, it becomes:(36)$$N_{sso}^{r}>\frac{1}{R_{\tau }}\left[{\left(\frac{1}{\varepsilon \cdot \exp\left[-{\left(r_{w}\right)}^{2}\right]}\right)}^{\frac{2}{3}}-1\right]$$

For $N_{ssmin}^{r}\;and\;N_{ssmax}^{r}$, with the approximation of $\mathrm{erf}\left(X\cdot r_{w}\right)\approx \frac{2}{\sqrt{\pi }}X\cdot r_{w}$,
(37)$$\begin{array}{c} N_{ssmin}^{r}>\frac{1}{R_{\tau }}\left[{\left(\frac{2}{R_{\tau }\cdot \varepsilon \cdot T_{min}\left(r_{w},\infty \right)}\right)}^{2}-1\right] \end{array}$$
(38)$$\begin{array}{c} N_{ssmax}^{r}>\frac{1}{R_{\tau }}\left[{\left(\frac{2}{R_{\tau }\cdot \varepsilon \cdot T_{max}\left(r_{w},\infty \right)}\right)}^{2}-1\right] \end{array}$$

The behavior of $N_{ss}^{r}$ according to $R_{\tau }$ for $r_{w}$ = 0 has already been shown in Figure 5, with an overall increase. We have plotted *N_sso_* and *N_ssmax_*, according to $r_{w}$, for $R_{\tau }=10$ as an example, as shown in Figure 9a, and the plot of the related time for reaching the steady state (using *N_ssmax_* and with diffusion time 0.28 µs) in Figure 9b.

From Figure 9, we can see that as $r_{w}$ increases, it takes more pulses (three orders of magnitude more) and this corresponds to a longer time to reach the steady state. Therefore, in reality, even though at the exact center of the beam, the temperature is stable, the periphery is still evolving. In particular, in the case of a moving beam, the maximum speed of scanning is limited by the change at the focus periphery increasing from zero.

## 4. The Mean Temperature in the Period between Two Pulses

For many transformations induced by laser irradiation (fictive temperature, crystallization, erasure of previously induced structures, stress relaxation, and so on), the large temperatures occurring within a pulse period are so brief that the system has no time to significantly respond. On the contrary, for smaller temperatures occurring at the end of the period, the system may have time to respond if the temperatures are not too small (this is the case for overlapping pulse contribution, i.e., heat accumulation). Therefore, the system responds efficiently predominantly for intermediate temperatures in the main part of the period. On the other hand, when *R_τ_* is small (large pulse *RR* versus diffusion time), temperature oscillations are relatively small whatever the radius, or for large radius values whatever *R_τ_* values (see Figure 8), the temperature oscillation can be neglected. For these, the use of an average temperature is relevant. In any case, the average value can be a guide for following the temperature distribution in space and its evolution. Hence, this section is devoted to simple expressions of average temperature values in the function of material and laser parameters.

We define the averaging by $\bar{T}\left(r,N\right)=\frac{1}{\tau_{p}}\int_{\begin{array}{c} pulse\;period\\ at\;N \end{array}}T(r,t)dt$, and this gives:(39)$$\begin{array}{c} \bar{T}\left(r,N\right)=\frac{1}{\tau_{p}}\int_{\begin{array}{c} pulse\;period\\ at\;N \end{array}}T\left(r,t\right)dt \end{array}$$

N.B. due to software problem, the average temperature is sometimes quoted as $\bar{T}$ and sometimes *T_mean_*. They have the same meaning.

### 4.1. Temperature at the Center ($\bar{T}\left(0,N\right)$)



(40)
$$\begin{array}{c} \bar{T}\left(0,N\right)=\frac{1}{\tau_{p}}\int_{\frac{t}{\tau_{p}}=N-1}^{\frac{t}{\tau_{p}}=N}\sum_{n=0}^{N-1}\frac{1}{{\left[1+\left(\frac{t}{\tau_{p}}-n\right)\cdot R_{\tau }\right]}^{\frac{3}{2}}}dt \end{array}$$



The two summations can be permuted as they do not operate on the same variables and are independent. We obtain:(41)$$\bar{T}\left(0,N\right)=\frac{1}{\tau_{p}}\sum_{n=0}^{N-1}\int_{\frac{t}{\tau_{p}}=N-1}^{\frac{t}{\tau_{p}}=N}\frac{1}{{\left[1+(\frac{t}{\tau_{p}}-n)\cdot R_{\tau }\right]}^{3/2}}dt=\frac{1}{R\tau }\sum_{n=0}^{N-1}{\left[-\frac{2}{{\left[1+(\frac{t}{\tau_{p}}-n)\cdot R_{\tau }\right]}^{1/2}}\right]}_{\frac{t}{\tau_{p}}=N-1}^{\frac{t}{\tau_{p}}=N}=\frac{2}{R_{\tau }}\left(1-\frac{1}{\sqrt{1+N\cdot R_{\tau }}}\right)$$

This result is obtained without approximation. Then, when $N\gg 1/R_{\tau }$:(42)$$\begin{array}{c} \bar{T}\left(0,\infty \right)=\underset{N\gg 1/R_{\tau }}{\lim}\bar{T}\left(0,N\right)=\frac{2}{R_{\tau }}\; \end{array}$$

We note that here, the steady state temperature at the center will reach:(43)$$\begin{array}{c} \frac{2\cdot T_{00}}{R_{\tau }}=\frac{2A\cdot P}{\pi^{\frac{3}{2}}\kappa w} \end{array}$$

It is the same expression as for $T_{max}\left(0,\infty \right)$ or $T_{min}\left(0,\infty \right)$ for small $R_{\tau }$ values. We can note in Figure 10 that $T_{max}\left(0,\infty \right)$ and $T_{min}\left(0,\infty \right)$ approach $\bar{T}\left(0,\infty \right)$ when $R_{\tau }$ is decreasing. $T_{max}\left(0,\infty \right)$ goes to 1 and $T_{min}\left(0,\infty \right)$ goes to 0 when $R_{\tau }$ is large. From Figure 10, we can also find the heat accumulation bound already defined in Figure 5 (with *ε* = 0,03). It corresponds to $T_{min}\left(0,\infty \right)$ = 0.03 and $R_{\tau }=12$. On the other hand, when $T_{min}\left(0,\infty \right)$ departs from $T_{max}\left(0,\infty \right)$ by less than approximately 10%, we can admit that the average T is applicable, i.e., for $R_{\tau }$ smaller than 0.17 (purple circle). In this case, we can apply the simple expression Equation (43).

#### The Effective Number of Pulses for Reaching the Limit $\bar{T}\left(0,\infty \right)$
*(*$N_{ssm}^{0}$)

With the same definition as above, the number of pulses to reach $\bar{T}\left(0,\infty \right)$, i.e., $\frac{\left|\bar{T}\left(0,\infty \right)-\bar{T}\left(0,N\right)\right|}{\bar{T}\left(0,\infty \right)}<\varepsilon$, $N_{ssm}^{0}$ is obtained:(44)$$\begin{array}{c} N_{ssm}^{0}>\frac{1}{R_{\tau }}\left[{\left(\frac{1}{\varepsilon }\right)}^{2}-1\right] \end{array}$$

$N_{ssm}^{0}$ are compared to $N_{ssmax}^{0}$ and $N_{sso}^{0}$ in Figure 11. The steady state of the mean temperature is reached as *T_max_*.

Therefore, the time for reaching the steady state here is not *R_τ_* dependent: $\tau_{D}\left[{\left(\frac{1}{\varepsilon }\right)}^{2}-1\right]\approx \frac{\tau_{D}}{\varepsilon^{2}}$ = 1111$\tau_{D}$ when ε=0.03. It is the value of the maximum *t_ss_* for reaching a steady state when $R_{\tau }\to 0$ (Figure 6).

With these analytical expressions of temperature at the steady state at the center of the focus ${(T}_{osc},T_{min}\left(0,\infty \right),T_{max}\left(0,\infty \right),\bar{T}\left(0,\infty \right))$, and the needed number of pulses ($N_{sso}^{0},N_{ssmax}^{0},N_{ssm}^{0})$, we have a clear view of how parameter $R_{\tau }$ influences the thermal situation at the focus center. The problem is now to extend these results to any place out of the center.

### 4.2. Temperature out of the Focus Center ($\bar{T}\left(r,N\right)$)

With the definition $\bar{T}\left(r,N\right)=\frac{1}{\tau_{p}}\int_{\begin{array}{c} pulse\;period\\ at\;N \end{array}}T(r,t)dt$, we have the average temperature in a period as:(45)$$\bar{T}\left(r_{w},N\right)=\frac{1}{\tau_{p}}\int_{\frac{t}{\tau_{p}}=N-1}^{\frac{t}{\tau_{p}}=N}\sum_{n=0}^{N-1}\frac{1}{{\left[1+\left(\frac{t}{\tau_{p}}-n\right)\cdot R\tau \right]}^{\frac{3}{2}}}\cdot \exp\left[-\frac{{\left(r_{w}\right)}^{2}}{1+\left(\frac{t}{\tau_{p}}-n\right)\cdot R\tau }\right]\cdot dt=\frac{\sqrt{\pi }}{R\tau \cdot r_{w}}\cdot \left[\mathrm{erf}\left(r_{w}\right)-erf\left(\frac{r_{w}}{\sqrt{1+N\cdot R\tau }}\right)\right]\underset{N\gg \frac{1}{R\tau }}{\to }\frac{\sqrt{\pi }}{R\tau \cdot r_{w}}\mathrm{erf}\left(r_{w}\right)\;So\;\bar{T}\left(r_{w},\infty \right)=\frac{\sqrt{\pi }}{R\tau \cdot r_{w}}\mathrm{erf}\left(r_{w}\right)$$

This limit when $N\gg 1/R_{\tau }$ is shown in Figure 12 and compared to the Gaussian shape of *T_max_* when $R_{\tau }$ is large and when $R_{\tau }$ is small. When *T_max_*(*r*) is Gaussian for the first case, *T_max_*(*r*) has the same shape that *T_mean_* has for the second case. As the erf function tends to 1 (already for $r_{w}>2$), the function tends to be hyperbolic and thus decreases much slower than a Gaussian one (see Figure 12). The amplitude is $\frac{2}{R_{\tau }}$. It is inversely proportional to $R_{\tau }\;$whatever $R_{\tau }$ value.

Consistently with the previously calculated *T_max_* and *T_min_*, the *T_mean_* curve width is equal to the beam Gaussian for large $R_{\tau }$ values and to the curve limit given in Equation (45) and shown in Figure 8a which is wider.

#### The Effective Number of Pulses for Reaching the Limit of *T_mean_ (*$N_{ssm}^{r}$*)*

The effective number of pulses $N_{ssm}^{r}$ with the approximation $\mathrm{erf}\left(X\cdot r_{w}\right)\approx \frac{2}{\sqrt{\pi }}X\cdot r_{w}$ as $X\left(N,R\tau \right)=\frac{1}{\sqrt{1+N\cdot R\tau }}$ < 1, is solved to be:(46)$$N_{ssm}^{r}\left(r_{w}\right)>\frac{1}{R_{\tau }}\left[{\left(\frac{2}{R_{\tau }\cdot \varepsilon \cdot \bar{T}\left(r_{w},R_{\tau }\right)}\right)}^{2}-1\right]=\frac{1}{R_{\tau }}\left[{\left(\frac{2\cdot r_{w}}{\varepsilon \cdot \sqrt{\pi }\cdot erf\left(r_{w}\right)}\right)}^{2}-1\right]$$

From the expression above, we see that the periphery of the distribution is stabilized later than the center as we noticed already in the previous section.

## 5. Application Examples

To demonstrate the practical significance of the aforementioned calculations, we are discussing below several problems where we can apply these equations to analyze the temperature effects.

**Laser-induced crystallization.** It is known that for crystallizing a glass, it is necessary to control temperature and time in order to penetrate the crystallization domain [38]. A method for reaching it with a pulsed laser is described in [39]. It is shown that crystallization with a single pulse is possible from the solid state if the beam scanning speed is sufficiently low according to the nucleation time and crystallization growth rate. For a larger scanning speed, it is necessary to increase the pulse energy or the *RR* to reach the crystallization domain. This is for the formation of nanocrystals that are orientable with laser polarization. The decrease in the speed leads to the growth of the nanocrystals. In turn, crystallization is still possible if the speed is increased but the pulse energy should be increased. In such a way, the temperature overcomes the melting one during a time long enough in the pulse period and the material is melted in such a way that crystallization does not progress more after each pulse. From the calculations in this paper, the best method appears to be a high *RR* with moderate pulse energy in order to maintain *T* (control of *T_max_* and *T_min_*) around the crystallization temperature.

**Erasure process during laser writing.** In the case of pure silica, there is a first regime called type I for which the refractive index increases for pure silica glasses [40,41]. It is partly produced by a change in fictive temperature [42,43]. For that, the time for the temperature to decrease until a given value of temperature has to be larger than the relaxation time of the glass. This time is roughly defined by the cooling time which is itself defined by the moving spatial curve for one pulse [44]. Pulse energy can be adjusted consequently.

In the case of the materials in which type II transformation is achievable giving rise to a large birefringence based on self-organized nanograting (NG) and nanopores, there is a pulse-energy–*RR*–scanning-speed-related domain [45,46]. This domain is limited for large pulse energies depending on *RR*. In addition, for large *RR* values, the retardance decreases until it is no longer possible to write NGs. One of the hypotheses is the following: NG is based on the existence of nanopores distributed in a self-organized NG [9]. Recent work [47] shows that the thermal stability of such an object is defined by the viscosity that itself depends on *T*. Therefore, as *T* increases when pulse energy and RR are increased, they have to be limited to avoid an in-pulse erasure after creation during the pulse.

**Concurrent processes.** In organic materials, according to pulse energy and RR for the same mean power, two different processes are observed whereas we could believe that modification is just dependent on mean power (i.e., dose) [48]. One is the destruction of the material at low *RR* and high *E_p_*, while the other is the creation of luminophores at high *RR* and low *E_p_*. We explain this by looking at the amplitude of the *T* oscillations; we can say that for the first case, the oscillations are large whereas for the other case, they are small. In the first case, the temperature is overcoming the decomposition temperature of the material but not in the second case.

**Size of the crystallized trace.** In [49], we find a size of the heat-affected zone that is much larger for the glass called Silica-SrTiO_3_ than for Silica-LiNbO_3_ for comparable laser parameters. *R_τ_* for the first is 84 and for the second is 11. The first remark is that there is almost no heat accumulation (separated pulse contribution to the temperature) especially for the STS glass. For this glass, it is even possible to use the formula for one contribution. Then, the size is defined by the lowest maximal temperature according to the radius (Equation (30)) at least larger than *T_g_*. Note that due to the expected size of the trace, the use of *T_mean_* is possible for intermediate $R_{\tau }$ values (Equation (45)).
$$\frac{\sqrt{\pi }}{R\tau \cdot r_{w}}\mathrm{erf}\left(r_{w}\right)=\bar{T}\left(r_{w},\infty \right)=\frac{T_{g}}{T_{00}}$$

**Crown effect.** In [49], we also find an example of a crystallized shell (the center is not crystallized). Similarly, as above, there is a highest maximal temperature that is equal to *T_melting_*. Above this temperature, the viscosity decreases strongly and other processes may appear (see [49]) to be blocking crystallization on cooling.

Speed effect on the laser trace width. In glasses for mid IR [50], the energy threshold for the appearance of a sudden spatial broadening depends on the scanning speed, so we can deduce that it is not related to temperature as the writing speed is not involved in the thermal diffusion for a speed lower than a few m/s.

There are also some remarks that we can deduce from the calculation:-If a process is actually independent of *RR*, it does not depend on temperature (see [46]);-The number of pulses received by the material locally depends on the scanning speed. As the number of pulses for reaching a steady state is different at the center than at the periphery, it is possible that the appearance of the trace on the edge depends on the scanning speed during the transient stage;-However, the transient stage is not dependent on the pulse energy.

## 6. Conclusions

This study derives analytical expressions for the temperature distribution at the steady state induced by ultrafast multi-pulses within a spherical geometry, based on the laser and constant material parameters. These expressions depend only on two parameters: the initial temperature at the center (denoted as *T*_00_) and a quantity *R_τ_*, defined as the ratio of the pulse period *τ_p_* to the diffusion time *τ_d_*. We recall that temperature oscillates between *T_max_* and *T_min_*, eventually reaching a steady state, and we calculate the minimum number of pulses required to attain this state. For ease of use, a geometry with spherical symmetry was chosen for the energy deposition in order to lead to simple temperature expressions compiled in Table 3. The Table 4 is a further simplification usable in most of the cases. This approach also facilitates a clear and precise definition of the onset of heat accumulation.

We analyzed the distribution of the temperature oscillations relative to the radius from the center and the parameter *R_τ_*. Oscillations are large at the center for large *R_τ_* values but decrease strongly for a large radius $r_{w}$ > 2, i.e., for the periphery where the light intensity decreases almost by a factor of 10. On the contrary, oscillations are minimal everywhere for small *R_τ_* values (i.e., high frequency or low thermal diffusivity). In such conditions, the average of the temperature from the last period can be used, yielding even simpler expressions. Additionally, we found that the periphery of the focus reaches the steady state later than the center. By examining the pulse number required for the steady state according to the radius, we can better control transformations in these regions and understand the variations from the center.

This work aids in understanding how temperature variations influence different experimental observations, mentioned at the end. It can also be helpful to detect if temperature is acting on the processes of direct laser writing.

Future work includes refining this approach by considering the asymmetry of fs focus, making differences between transversal radius and depth to deduce how the trace changes over time. Another interesting point is the T dependence of the physical–chemical parameters, but this cannot be investigated without finite element calculation. Our approach allows us to choose the most representative parameters for applying such a calculation. In addition, the asymmetry of the focal volume, along and perpendicular to the propagation axis, is a refinement of the present calculations that we could be interested in to include variations with the pulse energy (like Kerr focusing, plasma density defocusing, plasma mirror).

## Figures and Tables

**Figure 1 micromachines-15-00196-f001:**
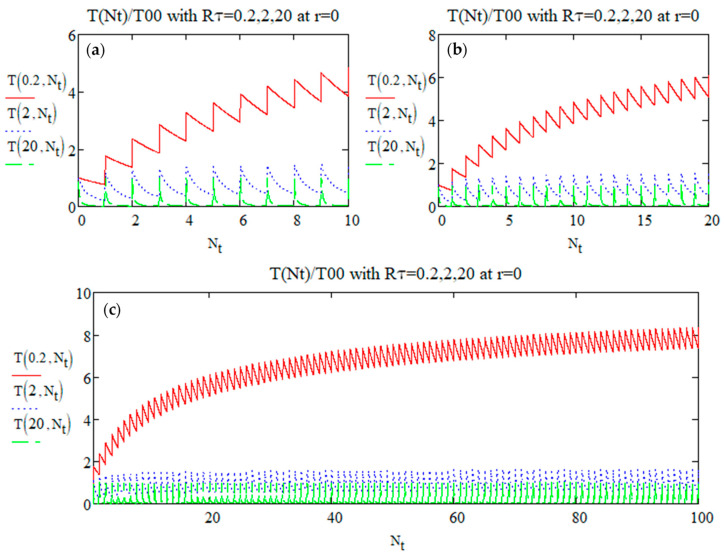
Plot of the relative temperature (Equation (10)) at the center *r_w_* = 0 according to the generalized pulse number $N_{t}=\frac{t}{\tau_{p}}$ with $R_{\tau }=0.2,\;2,\;20$ until several pulse numbers (**a**) 10, (**b**) 20, and (**c**) 100.

**Figure 2 micromachines-15-00196-f002:**
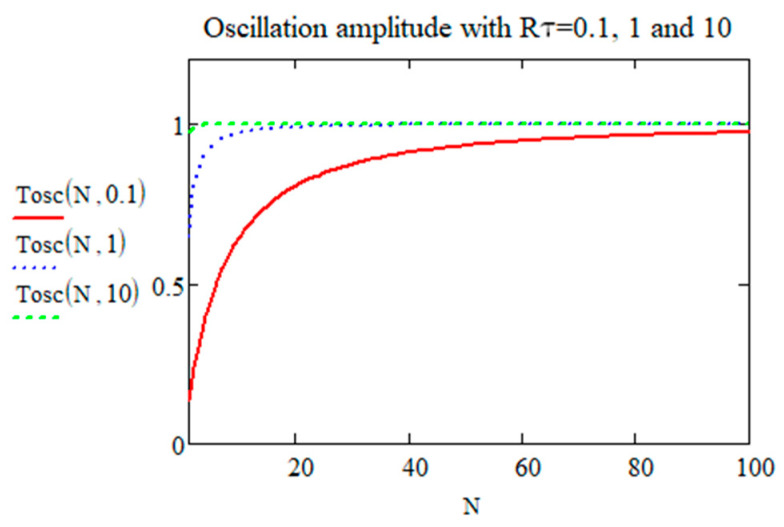
Temperature oscillation $T_{osc}\left(0,N\right)/T_{00}$ (here quoted *T_osc_*(*N*,*R_τ_*) dependence according to pulse number and for three values of *R_τ_* = 0.1, 1, and 10).

**Figure 3 micromachines-15-00196-f003:**
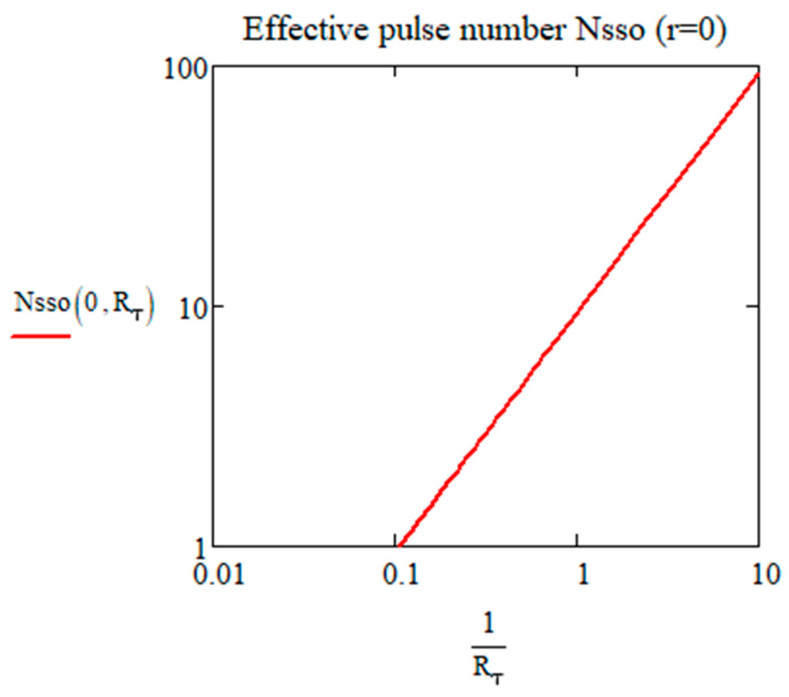
Number of pulses to reach the oscillation amplitude limit $N_{sso}^{0}$ according to $R_{\tau }$ from 0.1 to 100, with *ε* = 3%. For 1/*R_τ_* > 0.1, weak heat conduction, large *RR*, and vice versa.

**Figure 4 micromachines-15-00196-f004:**
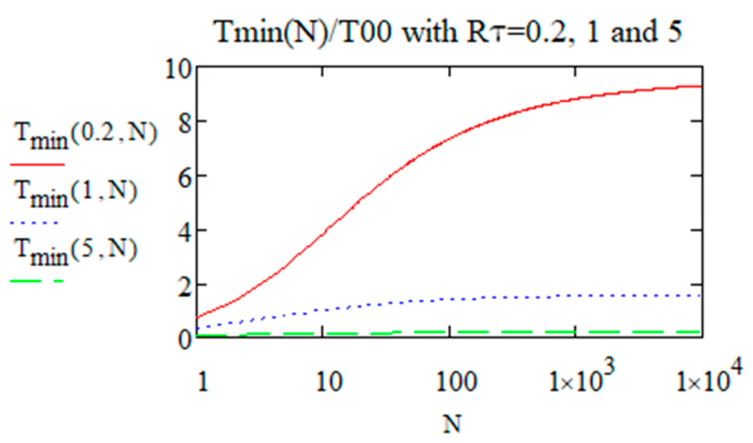
*T_min_/T*_00_ at *r_w_* = 0 according to *N* increasing from 1 to 10,000 when *R_τ_* = 0.2, 1, and 5.

**Figure 5 micromachines-15-00196-f005:**
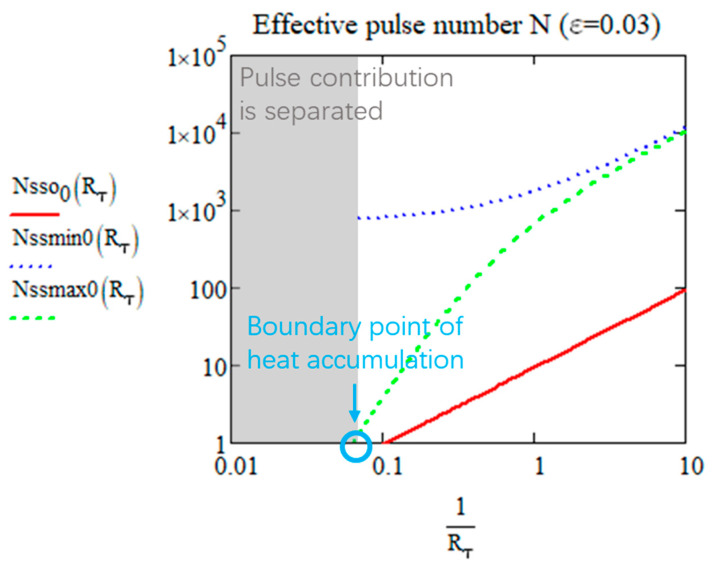
The effective number to reach the limit of *T_osc_*, *T_min_*, and *T_max_* according to $R_{\tau }$ from 0.1 to 100 for *ε* = 0.03. The boundary point is at *R_τ_* = 15.7.

**Figure 6 micromachines-15-00196-f006:**
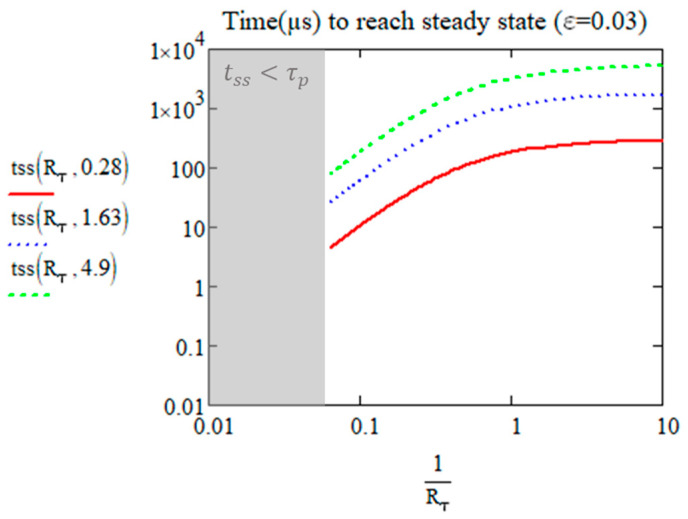
The time in µs to reach the steady state according to 1/$R_{\tau }$ value for glycine or silica (red), nifedipine (blue dash), and sucrose (green). The value of the second parameter in tss corresponds to *τ_d_* in Table 2.

**Figure 7 micromachines-15-00196-f007:**
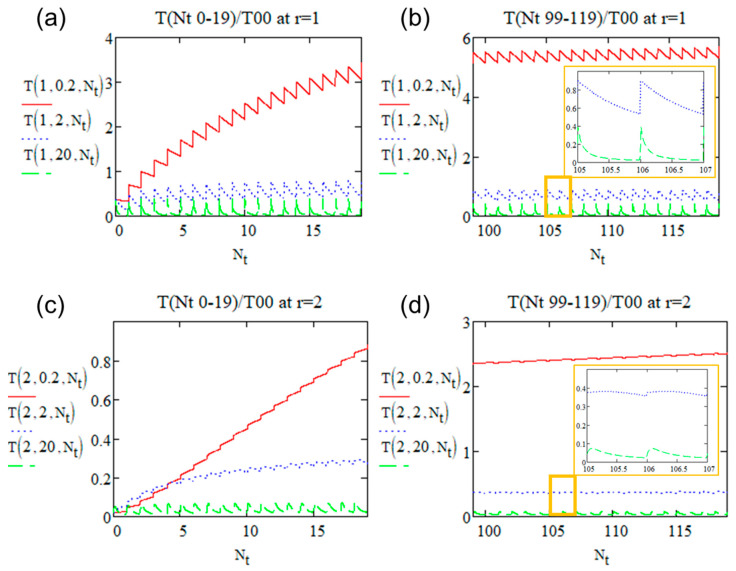
Plot of the relative temperature (*T/T*_00_) with $R_{\tau }$ = 0.2, 2, 20 at (**a**,**b**) *r = w* from (**a**) pulse 1 to pulse 20 and (**b**) pulse 100 to pulse 120 (**c**,**d**) *r = 2w* from (**c**) pulse 1 to pulse 20 and (**d**) pulse 100 to pulse 120. Inserts (**b**,**d**) zoom of pulse 106–108 of *R_τ_* = 2 and 20 at *r = w* and *r = 2 w*, respectively.

**Figure 8 micromachines-15-00196-f008:**
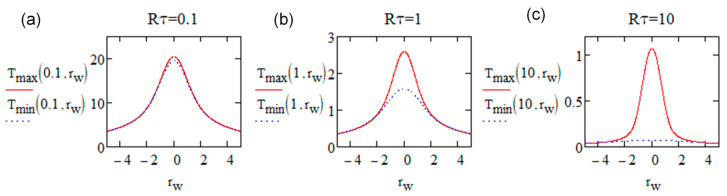
Spatial distribution of *T_min_* (blue dash, by Equation (29)) and *T_max_* (red, by Equations (27) and (30)) according to the relative radius *r_w_* when (**a**) $R_{\tau }=0.1$, (**b**) $R_{\tau }=1$, and (**c**) $R_{\tau }=10$.

**Figure 9 micromachines-15-00196-f009:**
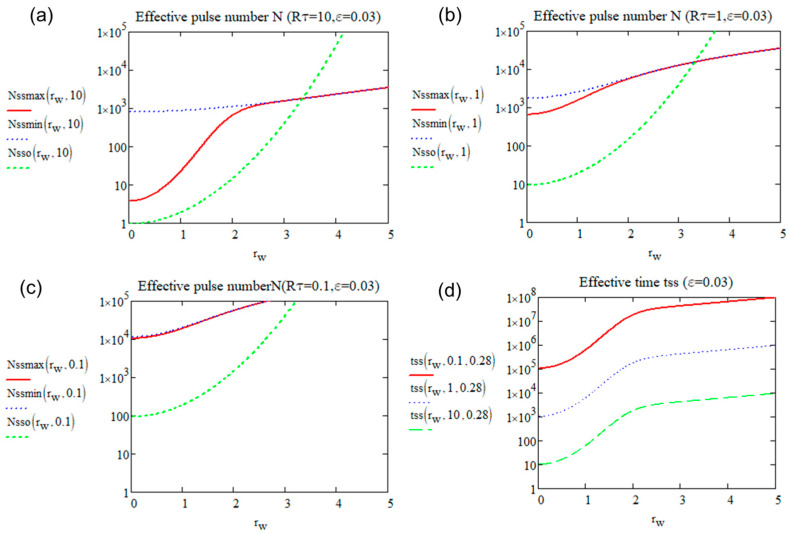
(**a**–**c**) Effective number of *N_ssmax_* (red), *N_ssmin_* (blue dash), and *N_sso_* (green) for reaching a steady state according to $r_{w}$ from 0 to 5 when (**a**) $\;R_{\tau }=10$; (**b**) $\;R_{\tau }=1$; (**c**) $R_{\tau }=0.1$. (**d**) The time (in μs) for reaching the steady state according to $r_{w}$ from 0 to 5 when$\;R_{\tau }=0.1,\;1$, and 10.

**Figure 10 micromachines-15-00196-f010:**
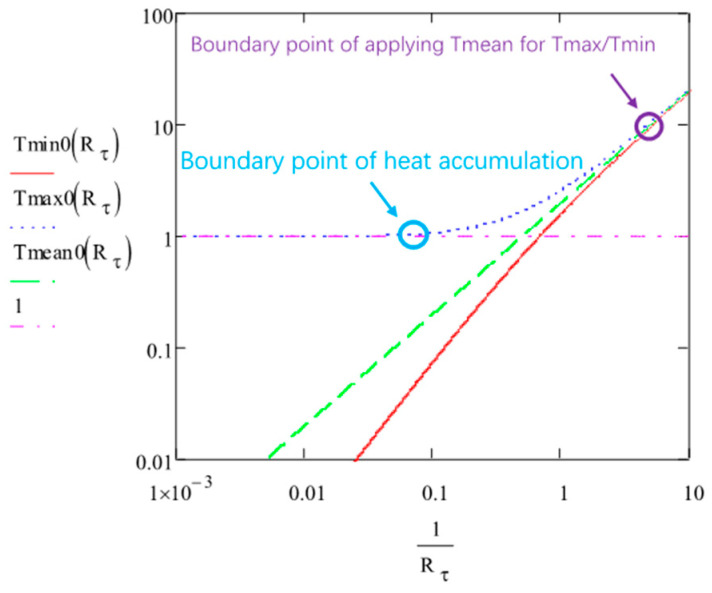
The plots of $T_{min}\left(0,\infty \right)$ (red), $T_{max}\left(0,\infty \right)$ (blue dash), and $\bar{T}\left(0,\infty \right)\;$(*T_mean_,* green dash) according to ${1/R}_{\tau }$. The defined boundary points of heat accumulation and negligible oscillation in the system are marked by a blue circle and purple circle, respectively (for definitions, see text).

**Figure 11 micromachines-15-00196-f011:**
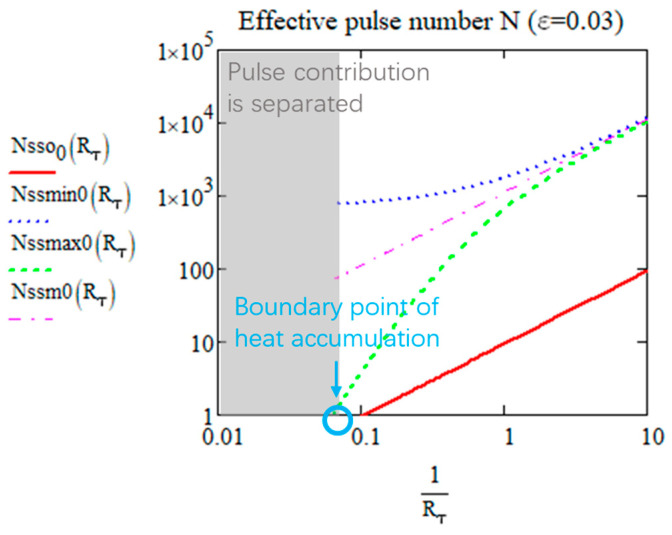
The effective number of pulses to reach the limit of *T_osc_* (red), *T_min_* (blue dash), *T_max_* (green dash), and *T_mean_* (pink dash) according to *R_τ_* from 0 to 5.

**Figure 12 micromachines-15-00196-f012:**
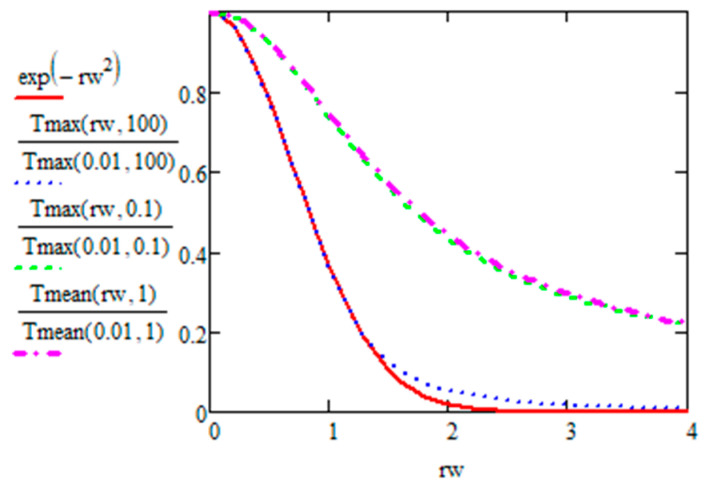
Plot of normalized $T_{max}\left(r_{w},R_{\tau }\right)$ and $\bar{T}\left(r_{w},R_{\tau }\right)$ at the steady state for $R_{\tau }=0.1$ and 100 for *T_max_* and 1 for *T_mean_*.

**Table 1 micromachines-15-00196-t001:** State of the art of the thermal simulation of laser–matter interactions.

Laser Type	Mode	Geometry	Source Shape	Solving Method	Refs.
CW	static	cylindrical	Gaussian(r) Beer–Lambert(z)	analytical	Lax [27]
pulsed	scanning	three axes	Gaussian(x,y) Beer–Lambert(z)	analytical	Sanders [28]
pulsed	scanning	three axes	Gaussian(x,y) Beer–Lambert(z)	analytical	Haba [29]
pulsed	static	spherical	Gaussian(r)	finite difference	Eaton [15] or Zhang [30]
pulsed	quasi-static	cylindrical	Gaussian(r) Beer–Lambert(z)	analytical one pulse	Sakakura[18]
CW	scanning	three axes	uniform deposition in parallelepiped volume	analytical	Miyamoto [31]
pulsed	static	cylindrical	Gaussian(r,z)	analytical	Miyamoto[25]
pulsed	static	cylindrical	Gaussian(r) Gaussian(z)	numerical	Shimizu [33]
pulsed	scanning	cylindrical	Gaussian(r) surface absorption	analytical	Rahaman [34,35]
pulsed	quasi-static	spherical	Gaussian(r)	analytical	this work

**Table 2 micromachines-15-00196-t002:** Pulse number needed (*N_ss_*) for reaching the steady state in materials, using Equation (23).

	SiO_2_	LNS	STS	Glycine	Zeonex	Sucrose	Nifedipine
$\tau_{d}$ (µs)	0.28	0.235	0.04	0.28	0.42	4.9	1.63
*RR* (kHz)	200	200	200	200	200	200	200
$R_{\tau }$	18	21	125	18	12	1	3
*N_ssmax_*	1	1	1	1	3	641	80

**Table 3 micromachines-15-00196-t003:** Analytical expressions of final temperature and the effective number for reaching steady state. *: condition of $r_{w},R_{\tau }$ (Equation (27), Figure A2).

	***r_w_*** **= 0**		
	$T\left(0,N\right)$	$T\left(0,N\to \infty \right)$	$N_{ss}^{0}\left(\varepsilon ,R_{\tau }\right)$
$T_{osc}\left(0,N\right)$	$1-X^{3}$	1	$\frac{1}{R_{\tau }}\left[{\left(\frac{1}{\varepsilon }\right)}^{2/3}-1\right]$
$T_{min}\left(0,N\right)$	$\frac{1}{2}\left[{X1}^{3}+X^{3}\right]+\frac{2}{R_{\tau }}\left[X1-X\right]$	$\frac{1}{2}{X1}^{3}+\frac{2}{R_{\tau }}X1$	$\frac{1}{R_{\tau }}\left[{\left(\frac{2}{R_{\tau }\cdot \varepsilon \cdot T_{min}\left(\infty \right)}\right)}^{2}-1\right]$
$T_{max}\left(0,N\right)$	1 + $T_{min}\left(N\right)$	1 + $T_{min}\left(\infty \right)$	$\frac{1}{R_{\tau }}\left[{\left(\frac{2}{R_{\tau }\cdot \varepsilon \cdot T_{max}\left(\infty \right)}\right)}^{2}-1\right]$
$\bar{T}\left(0,N\right)$	$\frac{2}{R_{\tau }}\left(1-X\right)$	$\frac{2}{R_{\tau }}$	$\frac{1}{R_{\tau }}\left[{\left(\frac{1}{\varepsilon }\right)}^{2}-1\right]$
	***r_w_* ≠ 0 ***		
	$T\left(r,N\right)$	$T\left(r,N\to \infty \right)$	$N_{ss}^{r}\left(\varepsilon ,R_{\tau }\right)$
$T_{osc}\left(r,N\right)$	$\exp\left[-{\left(r_{w}\right)}^{2}\right]-X^{3}\exp\left[-{\left({X\cdot r}_{w}\right)}^{2}\right]$	$\exp\left[-{\left(r_{w}\right)}^{2}\right]$	$\frac{1}{R_{\tau }}\left[{\left(\frac{1}{\varepsilon \cdot \exp\left[-{\left(r_{w}\right)}^{2}\right]}\right)}^{2/3}-1\right]$
$T_{min}\left(r,N\right)$	$\frac{1}{2}[{X1}^{3}\exp\left[-{\left({X1\cdot r}_{w}\right)}^{2}\right]\;+X^{3}\exp\left[-{\left({X\cdot r}_{w}\right)}^{2}\right]]\;+\frac{\sqrt{\pi }}{R_{\tau }\cdot r_{w}}[erf(X1\;\cdot r_{w}-erf\left({X\cdot r}_{w}\right)]$	$\frac{1}{2}{X1}^{3}\exp\left[-{\left({X1\cdot r}_{w}\right)}^{2}\right]\;+\frac{\sqrt{\pi }}{R_{\tau }\cdot r_{w}}erf\left({X1\cdot r}_{w}\right)$	$\frac{1}{R_{\tau }}\left[{\left(\frac{2}{R_{\tau }\cdot \varepsilon \cdot T_{min}\left(\infty \right)}\right)}^{2}-1\right]$
$T_{max}\left(r,N\right)$	$\exp\left[-{\left(r_{w}\right)}^{2}\right]+T_{min}\left(N\right)$	$\exp\left[-{\left(r_{w}\right)}^{2}\right]$+$T_{min}\left(\infty \right)$	$\frac{1}{R_{\tau }}\left[{\left(\frac{2}{R_{\tau }\cdot \varepsilon \cdot T_{max}\left(\infty \right)}\right)}^{2}-1\right]$
$\bar{T}\left(r,N\right)$	$\frac{\sqrt{\pi }}{R_{\tau }\cdot r_{w}}\cdot \left[\mathrm{erf}\left(r_{w}\right)-erf\left({X\cdot r}_{w}\right)\right]$	$\frac{\sqrt{\pi }}{R_{\tau }\cdot r_{w}}\mathrm{erf}\left(r_{w}\right)$	$\frac{1}{R_{\tau }\tau }\left[{\left(\frac{2\cdot r_{w}}{\varepsilon \sqrt{\pi }\cdot erf(r_{w})}\right)}^{2}-1\right]$

**Table 4 micromachines-15-00196-t004:** Practical approximated analytical expressions of final temperature and the effective number for reaching steady state. With $X\left(N,R_{\tau }\right)=\frac{1}{\sqrt{1+N\cdot R_{\tau }}}$ and $X1\left(R_{\tau }\right)=\frac{1}{\sqrt{1+R_{\tau }}}$. *: condition of $r_{w},R_{\tau }$ (Equation (27), Figure A2).

	*r_w_* = 0		*r_w_* ≠ 0 *	
	$T\left(0,N\to \infty \right)$	$N_{ss}^{0}\left(\varepsilon ,R_{\tau }\right)$	$T\left(r,N\to \infty \right)$	$N_{ss}^{r}\left(\varepsilon ,R_{\tau }\right)$
$T_{osc}\left(r,N\right)$	*1*	$\frac{1}{R_{\tau }}\left[{\left(\frac{1}{\varepsilon }\right)}^{2/3}-1\right]$	$\exp\left[-{\left(r_{w}\right)}^{2}\right]$ ***	$\frac{1}{R_{\tau }}\left[{\left(\frac{1}{\varepsilon \cdot \exp\left[-{\left(r_{w}\right)}^{2}\right]}\right)}^{2/3}-1\right]$
$T_{min}\left(r,N\right)$	$\frac{2}{R_{\tau }}X1$	$\frac{1}{R_{\tau }}\left[{\left(\frac{2}{R_{\tau }\cdot \varepsilon \cdot T_{min}\left(\infty \right)}\right)}^{2}-1\right]$	$\frac{\sqrt{\pi }}{R_{\tau }\cdot r_{w}}erf\left({X1\cdot r}_{w}\right)$	$\frac{1}{R_{\tau }}\left[{\left(\frac{2}{R_{\tau }\cdot \varepsilon \cdot T_{min}\left(\infty \right)}\right)}^{2}-1\right]$
$T_{max}\left(r,N\right)$	1 + $T_{min}\left(\infty \right)$	$\frac{1}{R_{\tau }}\left[{\left(\frac{2}{R_{\tau }\cdot \varepsilon \cdot T_{max}\left(\infty \right)}\right)}^{2}-1\right]$	$\exp\left[-{\left(r_{w}\right)}^{2}\right]$ *+* $T_{min}\left(\infty \right)$	$\frac{1}{R_{\tau }}\left[{\left(\frac{2}{R_{\tau }\cdot \varepsilon \cdot T_{max}\left(\infty \right)}\right)}^{2}-1\right]$
$\bar{T}\left(r,N\right)$	$\frac{2}{R_{\tau }}$	$\frac{1}{R_{\tau }}\left[{\left(\frac{1}{\varepsilon }\right)}^{2}-1\right]$	$\frac{\sqrt{\pi }}{R_{\tau }\cdot \;r_{w}}\mathrm{erf}\left(r_{w}\right)$	$\frac{1}{R_{\tau }}\left[{\left(\frac{2\cdot r_{w}}{\varepsilon \sqrt{\pi }\cdot erf(r_{w})}\right)}^{2}-1\right]$

## Data Availability

Data is contained within the article.

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
