# Peer review of "Usable Analytical Expressions for Temperature Distribution Induced by Ultrafast Laser Pulses in Dielectric Solids"

_micromachines, 2024, doi:10.3390/mi15020196_

Round 1

Reviewer 1 Report

Comments and Suggestions for Authors

Title: Usable Analytical Expressions for Spatial Distribution and Evolution Calculation of the Temperature Induced by Ultrafast Multi-Pulse Laser Irradiation in Solids

by Ruyue Que, Matthieu Lancry and Bertrand Poumellec

In the paper, the authors introduce an analytical model able to describe the spatial and temporal evolution of the temperature distribution induced by ultrafast laser pulses in dielectric medium when a spherical symmetry can be assumed. The model is based on the Fourier heat transfer equation and the laser is considered as a 3D spheric gaussian heat source. The state-of-the-art covers both the developed model and the main applications. The existing analytic models mainly concern cylindrical symmetry and/or Beer-Lambert(z) absorption low and are therefore not adapted to high focused beam with non-linear absorption. The thermal accumulation process at high repetition rate is studied. They demonstrate that it mainly depends on two parameters: (1) the ratio between the repetition period and the diffuse time; (2) the Initial temperature profile due to a single pulse absorption. Many useful analytical expressions are deduced concerning for example the steady state determination, the thermal oscillation during the accumulation, etc.

The assumptions are important and not always realist: the absorption volume Is assumed to be spherical (it is not often the case), the material properties do not depend on the temperature, there is no phase change, the laser spot is static (or quasi-static), the optical non-linear behavior of the medium is not considered. However, their objective is not to be better than a slow rigorous 3D finite element method, but to be able to find quickly good order of magnitude. For example: how many pulses, with which energy and which repetition rate to reach a given temperature in a given volume of a known material? What will be the temperature temporal stability? Etc.

It is a very interesting and useful work, that can be published after minor revisions.

1-The model is only adapted for dielectric solid. It must be written in the title.

2-Most of the assumptions are given. Some other must be added:
No material phase change is considered in the model.
No non-linear optical effects are taken into account. These effects often make move the focal point during the accumulation process.

3-I suggest pointing out that the typical application of this model is the thermal accumulation of high focused beam in material with non-linear absorption. Namely, the cylindrical symmetry and the Beer-Lambert law along z cannot be considered.

4-Table 3 and 4 are not readable. A landscape format for this page will be better, to increase the column size.

Other minor comments:

-Line 153 “rho and Cp is the…” must be replaced by “rho and Cp are the…”

-Line 155, the Laplace operator is not only given in spherical coordinate but also considering spherical symmetry. It must be modified.

-Line 160, the diffuse time expression (Dth/w) is wrong. w is not defined.

-Line 202, authors talk about “averaging on the pulse period”. Its is “on the pulse repetition period”.

-Line 221, authors talk about the temperature oscillation “in each period”. It would be clearer to say “between two pulses”.

-Line 268, “r=0” is true, but “r_w=0” would be clearer.

-Lines 327 and 328, the sentence is not clear. I propose: “increasing the pulse RR with constant Ep leads to faster temperature increase but NOT with constant average power”.

-Line 591, equation (45), the = of the first line must be aligned with the = of the second line.

-Line 677, in the conclusion, the authors say that their model is based on “material parameter”, it must be replaced by “constant material parameter”, due to these properties do not depend on the temperature.

-Line 681, in the conclusion, in the sentence “we chose a geometry of the energy deposition in order to lead to simple expressions”, “geometry” must be replaced by “geometry with spherical symmetry”.

-In the last paragraph, the possibility in the future to take into account optical non-linearities that make move the focus point must also be evocated.

-There are many spaces missing between values and unit, also before and after “=”.

Comments on the Quality of English Language

No remark.

Author Response

Reviewer 1

R1: In the paper, the authors introduce an analytical model able to describe the spatial and temporal evolution of the temperature distribution induced by ultrafast laser pulses in dielectric medium when a spherical symmetry can be assumed.

Authors: indeed, we used spherical symmetry for simplicity as we are attached to draw physical conclusions.

R1: The model is based on the Fourier heat transfer equation and the laser is considered as a 3D spheric gaussian heat source. The state-of the- art covers both the developed model and the main applications. The existing analytic models mainly concern cylindrical symmetry and/or Beer-Lambert(z) absorption low and are therefore not adapted to high focused beam with non-linear absorption. The thermal accumulation process at high repetition rate is studied. They demonstrate that it mainly depends on two parameters: (1) the ratio between the repetition period and the diffuse time; (2) the Initial temperature profile due to a single pulse absorption. Many useful analytical expressions are deduced concerning for example the steady state determination, the thermal oscillation during the accumulation, etc.

The assumptions are important and not always realist: the absorption volume Is assumed to be spherical (it is not often the case), the material properties do not depend on the temperature, there is no phase change, the laser spot is static (or quasi-static), the optical non-linear behavior of the medium is not considered. However, their objective is not to be better than a slow rigorous 3D finite element method, but to be able to find quickly good order of magnitude. For example: how many pulses, with which energy and which repetition rate to reach a given temperature in a given volume of a known material? What will be the temperature temporal stability? Etc.

Authors:

  • About temperature dependence of material properties, there are examples like (Mahnicka-Goremikina, et al. Materials 2022, 15, 7935. https://doi.org/10.3390/ma15227935) showing that Dth is decreasing by a factor smaller than 2 to 3 and thus favors the heat accumulation for thousand degrees. About Cp, it increases by a factor 2-3 and thus decreases Too but does not change significantly the physical conclusion.
  • If phase change is reversible (heating and cooling), we expect a small effect. If not, formulas are useful before the occurrence. We add few words on that point in page 3.
  • We said in the text (page 20) that for speed smaller than m/s, the formulas are applicable.
  • About optical non-linear absorption mentioned in page 1, we forgot to say that after absorption of a small part of the pulse energy, the absorption becomes linear due to limitations of the electron plasma density (M. Lancry, et al., Physical Review B 2011 Vol. 84 Issue 24). We add this remark in page 3.

R1 : 1-The model is only adapted for dielectric solid. It must be written in the title.

Authors : The title has been modified.

R1: 2-Most of the assumptions are given. Some other must be added:
No material phase change is considered in the model.
No non-linear optical effects are taken into account. These effects often make move the focal point during the accumulation process.

Authors : we assumed that there is no source or sink terms in the starting formulation section other than the laser one, so no phase change. Also no change of the material parameters due to any subsequent reaction.

We add some remarks about non-linear absorption (see above). However, the focal volume is dependent on optical propagation (self focusing effect and plasma density for femtosecond laser), the light affected volume is elongated along the direction of propagation according to the incident pulse energy. This is not taken into account in the present version of the calculation as spherical symmetry of the source is assumed. We add few words on that page3.

R1 3-I suggest pointing out that the typical application of this model is the thermal accumulation of high focused beam in material with non-linear absorption. Namely, the cylindrical symmetry and the Beer-Lambert law along z cannot be considered.

Authors: as a matter of fact, we said in page 1 that our objective is not a precise calculation but rather to understand thermal effect change on laser parameters. Thermal accumulation is one aspect but “when reaching a steady state”, “how change the heat affected zone”, “how large is the thermal oscillation according to the radius” are other useful aspects that are meet also with laser source with other symmetry.

R1 4-Table 3 and 4 are not readable. A landscape format for this page will be better, to increase the column size. 

Authors: Done in manuscript, need editor to modify the template.

R1 Other minor comments:

-Line 153 “rho and Cp is the…” must be replaced by “rho and Cp are the…” Done.

-Line 155, the Laplace operator is not only given in spherical coordinate but also considering spherical symmetry. It must be modified. Done.

-Line 160, the diffuse time expression (Dth/w) is wrong. w is not defined. Done.

-Line 202, authors talk about “averaging on the pulse period”. Its is “on the pulse repetition period”. Done.

-Line 221, authors talk about the temperature oscillation “in each period”. It would be clearer to say “between two pulses” Done.

-Line 268, “r=0” is true, but “r_w=0” would be clearer. Done everywhere in the text.

-Lines 327 and 328, the sentence is not clear. I propose: “increasing the pulse RR with constant Ep leads to faster temperature increase but NOT with constant average power”. Done.

-Line 591, equation (45), the = of the first line must be aligned with the = of the second line. Done.

-Line 677, in the conclusion, the authors say that their model is based on “material parameter”, it must be replaced by “constant material parameter”, due to these properties do not depend on the temperature.  Done.

-Line 681, in the conclusion, in the sentence “we chose a geometry of the energy deposition in order to lead to simple expressions”, “geometry” must be replaced by “geometry with spherical symmetry”. Done.

-In the last paragraph, the possibility in the future to take into account optical non-linearities that make move the focus point must also be evocated. Done at the end of the conclusion.

-There are many spaces missing between values and unit, also before and after “=”. Done

Reviewer 2 Report

Comments and Suggestions for Authors

This paper provides usable analytical expressions for spatial distribution and evolution calculation of the temperature induced by ultrafast multi-pulse laser irradiation in solids. Below are some revision suggestions that should be addressed before acceptance.

1.      The title is too long, may be the Spatial Distribution and Evolution Calculation of the Temperature could be revised to Temperature Distribution?

2.      Therefore, analytical expressions according to time and distance from the center of the focus, of temperature induced by multi-pulses absorption for pulse duration much smaller than ns within spherical energy source, are given in this paper according to the laser and material parameters. The sentence is too long, and hard to follow. Also, the abstract section suggests providing the main conclusions or innovative aspects of the theoretical calculations.

3.      In the equation 1, why can the heat conduction equation be used to explain the interaction process between ultrafast laser and matter. How to consider electron temperature and lattice temperature?

4.      For the equations 1-5, the temperature seems to be independent of the pulse width of the laser. This is difficult to understand.

5.      In my opinion, I suggest that the author should focus more on the accuracy of the model solution, rather than the subsequent calculation and analysis of temperature. Can we consider deleting some of the research content and strengthening the introduction of the model.

6.      For example, how to verify the correctness of the model. Can there be an experiment?

7.      In addition, for the interaction between multi pulse ultrafast laser and matter, the shielding problem of laser energy will become very serious. This seems to have not been taken into account in the author's analysis. Will this have an impact on the accuracy of the model?

8.      May be the manuscript needs careful editing, paying particular attention to English grammar, spelling, and sentence structure.

Comments on the Quality of English Language

Moderate editing of English language required.

Author Response

We thank a lot the reviewers for their remarks that we address below.

Reviewer 2

  1. The title is too long, may be the Spatial Distribution and Evolution Calculation of the Temperature could be revised to Temperature Distribution?

The title has been modified.

  1. Therefore, analytical expressions according to time and distance from the center of the focus, of temperature induced by multi-pulses absorption for pulse duration much smaller than ns within spherical energy source, are given in this paper according to the laser and material parameters. The sentence is too long, and hard to follow. Also, the abstract section suggests providing the main conclusions or innovative aspects of the theoretical calculations.

The abstract has been modified mainly at the beginning.

  1. In the equation 1, why can the heat conduction equation be used to explain the interaction process between ultrafast laser and matter. How to consider electron temperature and lattice temperature?

In the scenario of ultrafast laser interacting with matter, the laser energy is first absorbed by the electrons of the material, leading to an increase in electron temperature. Subsequently, these excited electrons transfer energy to the lattice through electron-lattice interactions, resulting in an increase in lattice temperature. Initially, these two systems are out of equilibrium, but over time (less than a few ps), they reach a state of thermal equilibrium. In this state of thermal equilibrium, the electron temperature and the lattice temperature are the same, then thermal diffusion occurs much latter. As the time scales are very different, we are allowed not to consider the details of this process and consider initial T(r) defined by the deposited laser energy (equation 2).

  1. For the equations 1-5, the temperature seems to be independent of the pulse width of the laser. This is difficult to understand.

When the time scales between the pulse width and the diffusion time are very different, the pulse width disappears in the integration of the equation 2.

  1. In my opinion, I suggest that the author should focus more on the accuracy of the model solution, rather than the subsequent calculation and analysis of temperature. Can we consider deleting some of the research content and strengthening the introduction of the model.

As we said in the text, our objective is not the accuracy but a useful simplicity for understanding the links of main aspects of the modifications with the material and laser parameters: “when reaching a steady state”, “how change the heat affected zone”, “how large is the thermal oscillation according to the radius” etc.... As a matter of fact, there are many reasons for departing from our approximation but we have seen that they do not change physical conclusions. For this paper, the objective is not to be better than a slow rigorous 3D finite element method, but to be able to find quickly good order of magnitude.

  1. For example, how to verify the correctness of the model. Can there be an experiment?

It is a great idea for improving this paper. It will be adopted for our next papers. The objective of the current paper is to present simple equations that facilitate quick estimate of the temperature magnitude. Additionally, it aims to assess the extent of influence that laser and material parameters have on the temperature.

  1. In addition, for the interaction between multi pulse ultrafast laser and matter, the shielding problem of laser energy will become very serious. This seems to have not been taken into account in the author's analysis. Will this have an impact on the accuracy of the model?

Yes, the shielding problem such as plasma scattering/absorption portion is strongly influencing the calculated temperature as the fraction  appearing in  varies. This quantity must be adjusted with the experiment but it does not vary during the steady state that is the state we consider here.. That’s why we are not discussing the accuracy between the model and experiment in this paper, as it takes more work for future improving and adjusting the model. Nonetheless, this problem does not alter our analysis of the function’s behavior. Furthermore, by comparison of the experimental thermal behavior of the laser induced modifications and the one predicted by the equations of this paper, we will be able to progress on the origin of the modifications.

  1. May be the manuscript needs careful editing, paying particular attention to English grammar, spelling, and sentence structure.

Yes. Thank you.

All the changes we included are collected in a pdf file : micromachines-2778229 with track change-BP2

Thank you again to the reviewers that allowed to improve the paper.

Round 2

Reviewer 2 Report

Comments and Suggestions for Authors

All my queries have been replied.